# MoFO: Momentum-Filtered Optimizer for Mitigating Forgetting in LLM Fine-Tuning

## Abstract

Large language models (LLMs) have demonstrated remarkable capabilities across a wide range of tasks. Typically, an LLM is first pre-trained on large corpora and subsequently fine-tuned on task-specific datasets. However, during fine-tuning, LLMs may forget some knowledge acquired in the pre-training stage, leading to a decline in general capabilities. To address this challenge, we propose a new fine-tuning algorithm termed Momentum-Filtered Optimizer (MoFO). As an extension of greedy block coordinate descent (BCD) methods, MoFO iteratively selects and updates the model parameters with the largest momentum magnitudes. MoFO achieves similar fine-tuning performance to the default fine-tuning algorithm while effectively mitigating knowledge forgetting. Furthermore, MoFO does not require access to pre-training data, making it highly suitable for scenarios where the pre-training data is unavailable, such as fine-tuning checkpoint-only open-source LLMs. We validate MoFO through rigorous convergence analysis and extensive experiments, demonstrating its superiority over existing methods in mitigating forgetting.

## 1 Introduction

The success of large language models (LLMs) lies in their strong capabilities in language understanding and generation. Typically, LLMs are initially pre-trained on extensive corpora to acquire general capabilities, and subsequently, they are fine-tuned on smaller, task-specific datasets to adapt to particular tasks or domains (Dai & Le, 2015; Kenton & Toutanova, 2019; Radford et al., 2018). However, it has been observed that during the fine-tuning process, LLMs may forget the knowledge acquired in pre-training, leading to a decline in general capabilities (Lin et al., 2023; Chen et al., 2020; Dong et al., 2021; Korbak et al., 2022; Luo et al., 2023). Therefore, addressing the issue of forgetting during fine-tuning has become an important research direction for LLMs.

In the field of continual learning, mitigating forgetting has already been a central focus. Continual learning (Wang et al., 2024a) involves training models sequentially on different tasks, which is analogous to the process of pre-training followed by fine-tuning in LLMs. Both involve different stages of training and face the challenge of forgetting previously acquired knowledge when learning new information. To address the issue, *replay-based methods* (Rolnick et al., 2019; Wang et al., 2020; Ouyang et al., 2022) use a replay buffer to store and revisit past data, in order to reinforce prior knowledge while learning new information. In LLM training, some replay-based methods are also used to mitigate forgetting (Shi et al., 2024; Roziere et al., 2023; Huang et al., 2024). However, replay-based methods face some practical limitations in LLMs. First, access to the original pre-training data is often restricted or infeasible. Many open-source LLMs, such as the Llama series (Touvron et al., 2023), do not fully disclose their pre-training datasets. Second, even when pre-training data is available, incorporating it into the fine-tuning process can substantially increase computational and memory costs, as the model must process a much larger and more diverse dataset.

In continual learning, another class of methods involves modifying the optimization process of models to mitigate forgetting (Wang et al., 2024a). Most optimization-based methods do not require direct access to past data but still depend on information from previous tasks. Gradient projection methods, such as GEM (Lopez-Paz & Ranzato, 2017) and OGD (Farajtabar et al., 2020), rely on gradient information from previous tasks, while landscape-based methods like Adam-NSCL (Wang et al., 2021) depend on checkpoints of prior models. However, in the context of LLMs, storing gradients

or checkpoints requires substantial memory due to the large model size, introducing a significant overhead to the fine-tuning process.

In this work, we aim to develop a forgetting-mitigation optimization method that does not utilize past data. We adopt an important insight in continual learning: the closer to the previous model, the less forgetting occurs. What optimization method might move in small distance from the initial point? We notice that the classical block coordinate descent (BCD) method (Tseng, 2001) is a good candidate, since it updates only a subset of parameters at each iteration, thus is implicitly biased towards closer solutions. Nevertheless, incorporating BCD into LLM fine-tuning presents some challenges. This is primarily because Adam, the predominant optimizer for LLM training (Radford et al., 2018; Zhang et al., 2024c), differs substantially from SGD studied in earlier continual learning works. It complicates both optimizer design and convergence analysis. Consequently, combining BCD with Adam is not a straightforward task.

To resolve the above challenges, we proposed Momentum-Filtered Optimizer (MoFO), a new optimization algorithm that integrates Adam with BCD. To achieve less forgetting while maintaining good fine-tuning performance, MoFO selects the most effective parameters at each iteration—those with large momentum magnitudes for reducing the fine-tuning loss. MoFO only modifies the optimizer without the need for pretraining data or introducing additional memory costs, which helps achieve its efficiency and effectiveness during fine-tuning. Our contributions are summarized as follows:

- We propose MoFO, a new training algorithm designed to mitigate the forgetting of pretraining knowledge during fine-tuning.
- We present a rigorous theoretical convergence result of the MoFO algorithm, providing a solid theoretical foundation that supports its good performance in fine-tuning tasks.
- We conduct experiments on various tasks, demonstrating that MoFO outperforms existing methods both in fine-tuning performance and mitigating forgetting.

## 2 RELATED WORKS

Catastrophic forgetting, a significant issue where models forget previously learned information upon learning new data, has received considerable attention in machine learning (McCloskey & Cohen, 1989; Goodfellow et al., 2013; Kemker et al., 2018; Ramasesh et al., 2021; Liu et al., 2024). We identify 5 primary categories of methods, as listed below.

**Replay-based methods**. These methods leverage past experiences to facilitate the learning of new tasks. The most classical scheme is experience replay, which involves replaying data of past tasks during incremental training (Rolnick et al., 2019) (Aljundi et al., 2019a; Hayes et al., 2019; Cha et al., 2021; Chaudhry et al., 2019b; Riemer et al., 2019b). Other variants utilize gradient information from old tasks (Lopez-Paz & Ranzato, 2017; Riemer et al., 2019a; Chaudhry et al., 2019a; Farajtabar et al., 2020; Aljundi et al., 2019b; Chaudhry et al., 2021; Tiwari et al., 2022). In LLMs, Yin et al. (2023); Wang et al. (2024b); Ouyang et al. (2022) propose replay-based methods to mitigate forgetting. MoFO is orthogonal to replay-based methods and can be combined with replay strategies.

**Regularization-based methods**. These methods introduce constraints to the training process to preserve past knowledge, such as adding regularization to the loss functions (Kirkpatrick et al., 2017; Aljundi et al., 2018; Zenke et al., 2017; Li et al., 2018; Ritter et al., 2018; Kumar et al., 2023) or the embedding/output changes (Li & Hoiem, 2017; Rannen et al., 2017; Buzzega et al., 2020; Huang et al., 2021; Cha et al., 2020). However, Aljundi et al. (2018); Panda et al. (2024); Lesort et al. (2019); Wu et al. (2022) point out some limitations of regularization-based approaches. They may exhibit poor adaptability to new tasks in long sequential learning. Moreover, some regularization methods typically require partial information of past models (Kirkpatrick et al., 2017). In contrast, MoFO does not require information from past models. We note that MoFO is also orthogonal to regularization-based methods and their combination is an interesting future direction.

**Model merging methods**. These methods balance learning new knowledge and retaining old knowledge by merging the new and past models. One line of research focuses on model averaging, which interpolates between the weights of different LLMs (Wortsman et al., 2022a;b; Eeckt et al., 2022; Yadav et al., 2024; Wortsman et al., 2022b; Lin et al., 2023; 2024). Another line of research relies on the observation that task-specific knowledge largely resides in a subspace of the weight space (Ilharco et al., 2023; Panigrahi et al., 2023; Gueta et al., 2023; Zhu et al., 2024), and leverage task

vectors or task localization to preserve pre-training knowledge in the fine-tuned models (Panigrahi et al., 2023; Yadav et al., 2024; Yu et al., 2024a).

**Architecture-based methods**. These methods modify the model's architecture in training. LoRA (Hu et al., 2022), as the most popular parameter-efficient fine-tuning (PEFT) method, freezes the pre-training weights and introduces low-rank trainable matrices. Variants of LoRA are applied in continual learning for LLMs (Ren et al., 2024; Wang et al., 2023a). However, LoRA is observed to forget less but also learn less than default fine-tuning (Biderman et al., 2024). Other approaches adaptively expand model capacity or isolate partial weights to mitigate interference between new and old tasks (Wang et al., 2023a; Razdaibiedina et al., 2023). In contrast, MoFO selects a subset of parameters to update at each iteration, but does not alter the total trainable parameters.

**Optimization-based methods**. These methods only modify the training algorithm to mitigate forgetting. Besides gradient projection (Wang et al., 2023a; Lopez-Paz & Ranzato, 2017) and the landscape-inspired methods (Wang et al., 2021) mentioned in the introduction, another popular type of optimization-based methods is **dynamic sparse training**, where training is restricted to partial parameters at each iteration. For instance, Hui et al. (2024) selectively freezes half of the model's parameters at each iteration. Ke et al. (2023b;a) introduce a soft-masking mechanism to regulate parameter updates based on their importance values. Further, Zhang et al. (2024a) combines selective module updating with soft-masking. MoFO falls in the realm of dynamic sparse training. Compared to existing optimization-based methods, MoFO relies on a simpler parameter selection mechanism based on momentum values only, thus not introducing much extra computation or design.

## 3 MOMENTUM FILTERED OPTIMIZER (MoFO)

### 3.1 MOTIVATION

**Correlation between Distance and Forgetting**

During fine-tuning, different training methods typically converge to distinct minima. Although all these minima achieve relatively small fine-tuning losses, their distances from the pre-trained model can vary significantly. To see this, we fine-tune Pythia-160M on a subset of the FLAN dataset[1] using Adam (Kingma & Ba, 2014) and Lion (Chen et al., 2024). As illustrated in Figure 1, Adam and Lion converge to distinct minima. Notably, while both optimizers achieve similar fine-tuning losses (Figure 1(a)), the minimum reached by Adam is much closer to the pre-trained model compared to Lion, being only about 20% of Lion's distance. See the calculation of distance in Appendix D.4.

Furthermore, we observe that the extent of forgetting during fine-tuning is correlated with the distance from the pre-trained model. As shown in Figure 1(b), the model fine-tuned using Adam remains closer to the pre-trained model and exhibits a smaller increase in pre-training loss. Additionally, as illustrated in Figure 1, fine-tuning with Adam results in less forgetting (measured by commonsense reasoning) than fine-tuning with Lion. To test the generality of this observation, we conduct a larger-scale exploratory experiment. We fine-tune a larger model, LLaMA2-7B, on the MetaMathQA dataset using three optimizers: Adam, Lion, and our proposed MoFO optimizer (to be introduced in Section D.4). To achieve varying distances from the pre-trained model, we fine-tune each model for 0.5, 1, 1.5, and 2 epochs (309 steps per epoch). Details are provided in Appendix D.3. Figure 2 demonstrates a strong positive correlation between distance and the increase in pre-training loss, as well as a strong negative correlation with accuracy on MMLU. These findings suggest that maintaining a model closer to its pre-trained state may help better preserve the pre-training knowledge.

**Selective Updates for Mitigating Forgetting**

Motivated by the strong correlation between forgetting and the distance from the pre-trained model, we seek to design an optimizer that encourages the fine-tuned model to keep closer to the pre-trained model. To achieve this, we draw inspiration from the classical block coordinate descent (BCD) method (Tseng, 2001), which updates only a subset of parameters during each iteration. We anticipate that by restricting updates to a subset of parameters—similar to the BCD approach—the overall adjustments from the pre-trained model will be smaller than those made by Adam, the default fine-tuning optimizer. thereby mitigating the forgetting of pretrained knowledge.

---

[1]The subset used is 'definite_pronoun_resolution_10templates,' available at `https://huggingface.co/datasets/Muennighoff/flan`. The learning rate is 2e-5 and the batch size is set as 64.

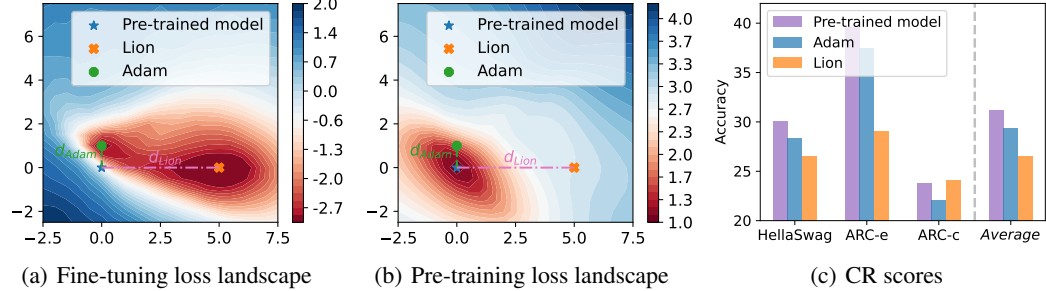

(a) Fine-tuning loss landscape     (b) Pre-training loss landscape     (c) CR scores

Figure 1: The loss landscapes of Pythia-160M after fine-tuning on a subset of the FLAN dataset using Adam and Lion. We plot the loss landscapes on (a) the fine-tuning dataset and (b) the pre-training dataset (Pile dataset (Gao et al., 2020)) and (c) the accuracies on CR tasks, including HellaSwag, ARC-c, and ARC-e. A logarithmic scale is applied to the loss values for better visualization. Two training methods converge to different minima with similar fine-tuning loss. Lion converges to a farther minimum from the pre-trained model and performs more forgetting than Adam.

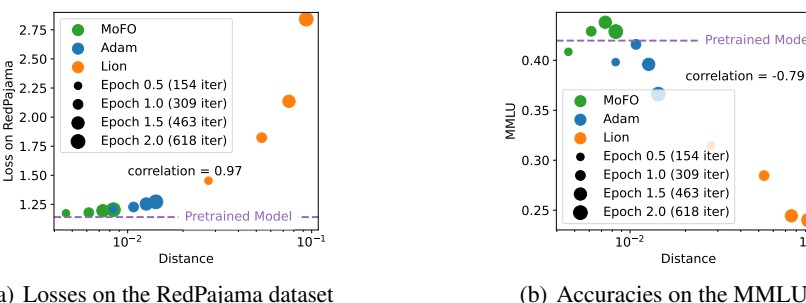

(a) Losses on the RedPajama dataset     (b) Accuracies on the MMLU

Figure 2: (a) Loss changes on the RedPajama dataset and (b) average accuracy changes on MMLU benchmark of Llama-2-7B after fine-tuning on MetaMathQA using Adam, Lion, and MoFO for 0.5, 1, 1.5, 2 epochs. Given that LLaMA2's original training data is not publicly available, we use the RedPajama dataset (Computer, 2023) as a comparable alternative. The results show a strong positive correlation between the distance from the pre-trained model and the extent of forgetting.

A key challenge is how to design such a selective updating strategy that maintains competitive performance to the default fine-tuning. One idea is to follow the Gauss-Southwell rule used in the coordinate descent method (Nutini et al., 2015). This rule selects parameters with the largest gradients, which are likely to contribute the most to reducing the loss in each iteration. However, for the default fine-tuning optimizer Adam, updates are influenced more by the momentum term. Therefore, we propose to modify the Adam optimizer to update only the parameters with the *largest momentum magnitudes*. By focusing on the most significant updates, our method, which we call the MoFO optimizer, aims to fine-tune models effectively while maintaining closer to their pre-trained state. We will discuss the details and formulation of MoFO in the next section.

## 3.2 ALGORITHM FORMULATION

We formally introduce the Momentum-Filtered Optimizer (MoFO) in Algorithm 1. First, all model parameters are partitioned into $B$ blocks. At each iteration, MoFO first computes the gradient and momentum terms for parameters in each block following the standard rule of Adam, as shown in Lines 5-9. Then, MoFO selects and updates the parameter entries with the largest $\alpha\%$ momentum magnitudes in each parameter block, as shown in Lines 10-13, where the update fraction $\alpha\%$ is a pre-determined hyperparameter. This momentum filtering mechanism is illustrated in Figure 3.

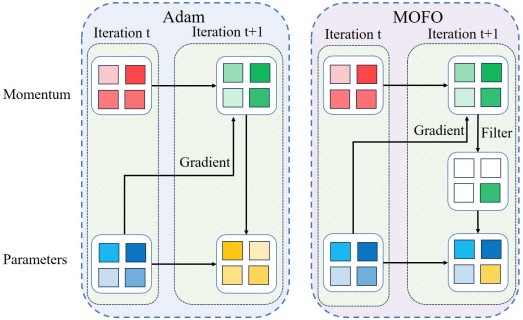

Figure 3: Illustration of MoFO.

---

**Algorithm 1** Momentum Filtered Optimizer (MoFO)

---

1: Input: Filtering threshold $\alpha\%$, number of partitions $B$ with the $k$-th partition of size $d_k$, hyperparameters $\beta_1, \beta_2$ of Adam optimizer, learning rate schedule $\{\eta_t\}$.
2: Initialize $m_0, v_0$ as zero tensors.
3: **for** iteration $t$ from $1, 2, \ldots$ until converge **do**
4:     **for** partition $k$ from $1$ to $B$ **do**
5:         $g_t^{(k)} = \nabla_{(k)} \mathcal{L}_{finetune}(\theta_{t-1})$
6:         $m_t^{(k)} = \beta_1 m_{t-1}^{(k)} + (1 - \beta_1) g_t^{(k)}$
7:         $v_t^{(k)} = \beta_2 v_{t-1}^{(k)} + (1 - \beta_2) g_t^{(k)} \circ g_t^{(k)}$
8:         $\hat{m}_t^{(k)} = m_t^{(k)} / (1 - \beta_1^t)$
9:         $\hat{v}_t^{(k)} = v_t^{(k)} / (1 - \beta_2^t)$
10:         **for** entry index $i$ from $1$ to $d_k$ **do**
11:             $[\mathtt{FLT}_\alpha^{(k)}(m_t)]_i = 1$ **if** $|(m_t^{(k)})_i|$ is within the top-$\alpha\%$ of $|m_t^{(k)}|$'s values **else** $0$
12:         **end for**
13:         $\theta_t^{(k)} = \theta_{t-1}^{(k)} - \eta_t \cdot (\hat{m}_t^{(k)} \odot \mathtt{FLT}_\alpha^{(k)}(m_t)) / \sqrt{\hat{v}_t^{(k)}}$       # Momentum Filtering
14:     **end for**
15:     $\theta_t = \mathtt{Concat}(\theta_t^{(1)}, \ldots, \theta_t^{(B)})$
16: **end for**

---

Mathematically, the filter can be represented as follows. Consider a momentum vector $m = (m^{(1)}, \ldots, m^{(B)})$, where each $m^{(k)} \in \mathbb{R}^{d_k}$ corresponds to the $k$-th block of parameters with dimensionality $d_k$. The top-$\alpha\%$ filter, denoted as $\mathtt{FLT}_\alpha(m)$, is defined as $\mathtt{FLT}_\alpha(m) = (\mathtt{FLT}_\alpha^{(1)}(m), \ldots, \mathtt{FLT}_\alpha^{(B)}(m))$, where the $i$-th entry of $\mathtt{FLT}_\alpha^{(k)}(m)$ is given by

$$\left[ \mathtt{FLT}_\alpha^{(k)}(m) \right]_i = \begin{cases} 1 & \text{if } |m_i^{(k)}| \text{ is within the top-}\alpha\% \text{ of } |m^{(k)}| \text{ values,} \\ 0 & \text{otherwise,} \end{cases} \quad (1)$$

for $i = 1, 2, \cdots, d_k$, $k = 1, 2, \cdots, B$. In our Momentum-Filtered Optimizer (MoFO), this filter $\mathtt{FLT}_\alpha$ is applied to the momentum $m_t$, selecting the entries with the largest magnitudes for updating.

For the parameter partitioning, we note that the network architecture is naturally composed of different modules (e.g., weight matrices, and bias terms). In the PyTorch implementation, the parameters of different modules (along with their gradients and momenta) are naturally stored in separate data tensors. Therefore, we adopt the default partitioning of model parameters as implemented in PyTorch. This allows us to select and update the top $\alpha\%$ parameters in each block without introducing much implementation overhead. See detailed explanation of the partitioning in Appendix D.4.

MoFO efficiently selects and updates the most "influential" parameters, as dictated by the momentum's magnitude. We will later show that this strategy alleviates the forgetting of pre-training knowledge. Further, we argue that filtering the momentum is more effective than filtering the gradient. In Section 4.4, we will empirically demonstrate that MoFO's momentum-based filtering rule outperforms other filtering rules in fine-tuning tasks. This improvement might be attributed to the fact that momentum provides a more stable and accumulated estimate of parameter importance compared to gradients, which are inherently noisier and more prone to fluctuations.

### 3.3 CONVERGENCE RESULT

In this section, we present the convergence result of MoFO for non-convex loss functions. For the simplicity of analysis, we consider the full-batch version of MoFO, with hyperparameters satisfying the following assumption.

**Assumption 1.** *The first and second order momentum hyperparameters $\beta_1$ and $\beta_2$ satisfy $0 < \beta_1 < \sqrt{\beta_2} < 1$. The learning rate schedule at step $t$ is $\eta_t = \eta/\sqrt{t}$ for some $\eta > 0$.*

**Theorem 1** (Convergence of MoFO). *Suppose that the loss function $\mathcal{L}$ is lower bounded by $\mathcal{L}^*$ and the gradient $\nabla \mathcal{L}$ is Lipschitz continuous with constant $L$. For the MoFO with hyperparameters satisfying Assumption 1, it holds that*

$$\min_{0 \le t \le T-1} \|\nabla\mathcal{L}(\theta_t)\|_\infty = \min_{1 \le t \le T} \|g_t\|_\infty = \mathcal{O}\left(\frac{\log T}{\sqrt{T}}\right) \quad as\ T \to \infty.$$

Although MoFO is designed to mitigate forgetting by updating only a small subset of parameters at each step it is guaranteed to converge to a critical point of the fine-tuning loss function under theoretical assumptions of bounded gradient and Lipschitz smoothness. This result provides theoretical evidence that MoFO can achieve competitive performance in fine-tuning tasks.

Our proof is inspired by the convergence analysis of full-batch Adam from Shi et al. (2021). A pivotal step in their proof involves applying the descent lemma to Adam, resulting in:

$$\mathcal{L}(\theta_t) - \mathcal{L}(\theta_{t-1}) \le -\frac{\eta}{\sqrt{t}} \sum_{i=1}^{d} g_{i,t} \frac{\hat{m}_{i,t}}{\sqrt{\hat{v}_{i,t}}} + \frac{L}{2}\|\theta_t - \theta_{t-1}\|_2^2. \tag{2}$$

Through a series of manipulations, they derive:

$$C_1\|g_t\|_1/\sqrt{t} \le \mathcal{L}(\theta_{t-1}) - \mathcal{L}(\theta_t) + C_2/t. \tag{3}$$

Summing this inequality from $t = 1$ to $T$ yields the convergence result for Adam in terms of a diminishing $\ell_1$-norm of gradient, given by

$$\min_{1 \le t \le T} \|g_t\|_1 = \mathcal{O}\left(\frac{\log T}{\sqrt{T}}\right). \tag{4}$$

Stemming from BCD methods, MoFO updates only a subset of parameters at each iteration. Consequently, the summation over all coordinates $i$ in (2) reduces to a summation over the coordinates selected by the momentum filter. Thus, instead of (3), we derive the following inequality:

$$C_1\|g_t \odot \mathtt{FLT}_\alpha(m_t)\|_1/\sqrt{t} \le \mathcal{L}(\theta_{t-1}) - \mathcal{L}(\theta_t) + C_2/t. \tag{3'}$$

We note that the update in MoFO is filtered based on the momentum magnitude rather than the gradient magnitude. This introduces a non-trivial challenge because the left-hand side of (3') being small does not necessarily imply that the gradient norm is small. This complicates the analysis, and we cannot directly establish a convergence similar to (4).

By carefully analyzing the interaction between momentum magnitudes and gradient norms, we show that the momentum-based filtering in MoFO introduces only a diminishing gap of $\mathcal{O}(1/\sqrt{t})$ between $\|g_t \odot \mathtt{FLT}_\alpha(m_t)\|_1$ and the $\ell_\infty$-norm of the gradient, $\|g_t\|_\infty$. The gap is small enough to ensure that the overall convergence rate does not degrade compared to Adam. As a result, we establish a convergence result for MoFO in terms of $\|g_t\|_\infty$, presented in Theorem 1.

We remark that the choice of $\beta_1$ and $\beta_2$ in Assumption 1 aligns with that used in analyzing full-batch Adam (Shi et al., 2021). Furthermore, the use of a diminishing learning rate in Theorem 1 is crucial for ensuring the stability of updates and avoiding divergence in the optimization process.

In summary, Theorem 1 demonstrates that despite updating only a subset of parameters, MoFO maintains the same convergence rate as Adam. This highlights the theoretical robustness of the momentum filter design in MoFO. We believe this result could provide valuable insights into adaptive optimization methods with filtering mechanisms.

# 4 EXPERIMENTS

## 4.1 EXPERIMENTAL SETTINGS

We verify the effectiveness of MoFO on **instruction fine-tuning** and **continual fine-tuning**. We use Llama-2-7B (Touvron et al., 2023), Gemma-2B-IT (Team et al., 2024), and TinyLlama-1.1B (Zhang et al., 2024b) as our base models. The instruction fine-tuning datasets cover question-answer pairs from different domains like mathematical reasoning and medical knowledge. Specifically, the datasets include: MetaMathQA (Yu et al., 2024b) and PMC-LLaMA-Instructions (Wu et al., 2024). We randomly sample 39.5K and 51K instances from these datasets, respectively, for training the LLMs. Additionally, We investigate the performance of MoFO in the continual fine-tuning scenario by implementing our approach on the TRACE benchmark dataset (Wang et al., 2023b).

**Evaluation metrics for instruction fine-tuning.** We employ widely used benchmarks to assess the performance and potential forgetting effects on the general capabilities of LLMs after instruction fine-tuning. These benchmarks include MMLU (Hendrycks et al., 2021) (0-shot) for factual knowledge;

ARC-Challenge, ARC-Easy (Clark et al., 2018), and HellaSwag (Zellers et al., 2019) (0-shot) for commonsense reasoning (CR); GSM8K (Cobbe et al., 2021) (5-shot) for mathematical reasoning; HumanEval (Chen et al., 2021) (pass@10) for code generation; PubMedQA (Jin et al., 2019), MedMCQA (Pal et al., 2022), and MedQA (Jin et al., 2021) (0-shot) for medical question answering (MedQ) [2]; IFEval (0-shot) for instruction following.

**Evaluation metrics for continual fine-tuning.** To evaluate the LLM's performance in continual learning, we consider two key metrics in this scenario: Overall Performance (OP) (Chaudhry et al., 2018) and BackWard Transfer (BWT) (Lopez-Paz & Ranzato, 2017).

For more descriptions and implementation details of these metrics and datasets, see Appendix D.

### 4.2 INSTRUCTION FINE-TUNING

In this section, we investigate the effectiveness of the MoFO algorithm in both preserving general capabilities and learning fine-tuning tasks. The implementation details are provided in Appendix D. The specific hyperparameter settings in each experiment are provided in Appendix D.3.

**LLM Fine-tuning strategy baselines.** We compare the proposed MoFO algorithm with the default fine-tuning approach and methods designed to mitigate forgetting. These baselines include: **Default fine-tuning (Default FT)** refers to the full-parameter fine-tuning approach using the Adam optimizer. (We note that due to the substantial memory requirements, SGD and low-memory implementation of Adam/SGD (e.g., LOMO (Lv et al., 2023)) has also been used in LLM fine-tuning. Thus, our experiments also include the full-parameter **SGD** for comparison. ) **Half Fine-tuning (HFT)** (Hui et al., 2024) randomly updates half of the parameter blocks within each transformer layer at each iteration while the other half are frozen. HFT can be considered a specific case of the BCD algorithm. **LoRA** (Hu et al., 2022) is a widely-used, parameter-efficient fine-tuning method. LoRA trains low-rank matrix adaptations on the base model's weights. Recent work (Biderman et al., 2024) demonstrates that LoRA can mitigate forgetting.

Table 1: The performance of the fine-tuning task (math), measured by GSM8K, and the general capability scores of Llama-2-7B after fine-tuning on the MetaMathQA dataset. The figure on the right visualizes both GSM8K accuracy and general capability scores. The results show that MoFO achieves comparable performance in the fine-tuning task, while significantly mitigating forgetting of general capabilities. Bold values denote the best results among these methods.

| Method | GSM8K | General Capability | | | |
| --- | --- | --- | --- | --- | --- |
| | | CR | MMLU | HumanEval | Avg. |
| Llama-2-7B | 13.7 | 65.6 | 42.0 | 24.2 | 43.9 |
| Default FT | 49.4 | 62.3 | 36.6 | 16.1 | 38.3 |
| SGD | 25.8 | 64.3 | 31.0 | 24.4 | 39.9 |
| HFT | 47.5 | 65.5 | 42.3 | 23.6 | 43.8 |
| LoRA | 43.3 | 65.1 | 37.7 | **26.4** | 43.1 |
| MoFO | 47.7 | **65.7** | **42.7** | 24.6 | **44.3** |

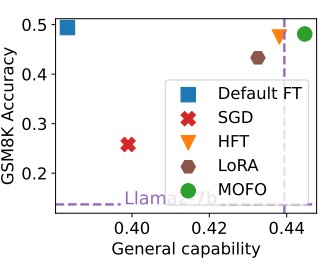

**Results of fine-tuning on MetaMathQA.** We fine-tune Llama-2-7B on MetaMathQA using various baseline methods and present the experimental results on mathematical reasoning (GSM8K) and general capabilities in Table 1. We report the experimental results of LoRA under the best-performing hyperparameter configuration on the fine-tuning task. These results demonstrate the effectiveness of our proposed MoFO algorithm in both optimization and mitigating forgetting.

MoFO is compatible to the performance of Default FT and HFT on the math task, yet significantly outperforms these methods in preserving general capability. Specifically, Default FT shows a decline of 5.4% in MMLU accuracy and HFT experiences a drop of 0.6% in HumanEval. In contrast, our MoFO not only maintains but slightly improves these general capability scores by an average of 0.4%. We also observe that MoFO significantly outperforms SGD in both forgetting mitigation and fine-tuning performance. Additionally, MoFO outperforms LoRA in fine-tuning performance, achieving a

---

[2]For CR and MedQ, we report the average of the benchmarks they comprise.

substantial $4.4\%$ improvement on the GSM8K benchmark. While LoRA suffers from forgetting of pre-trained capabilities, MoFO effectively preserves the general capabilities. An interesting finding is that LoRA demonstrates reduced forgetting of coding abilities. Although it does not match MoFO's performance in other aspects, LoRA remains a promising method deserving further investigation.

**Comparison from a Pareto perspective.** Generally, improving performance on the fine-tuning task and reducing forgetting are often a pair of competing objectives. It's intriguing to study how different fine-tuning methods balance this tradeoff. By adjusting the hyperparameters of different methods, we can observe a set of fine-tuned models, each representing a different tradeoff between fine-tuning performance and forgetting. The Pareto frontier formed by these models helps visualize the tradeoffs, and we can identify which method offers the best balance between fine-tuning and forgetting.

In this comparison, we also include traditional regularization methods such as $L_2$-regularization (Kirkpatrick et al., 2017) and $L_1$-regularization (Panigrahi et al., 2023), which are not specifically designed for large models. These methods modify the original fine-tuning loss $\mathcal{L}_{finetune}(\theta)$ by adding a regularization term. For $L_2$-regularization, the modified loss is $\mathcal{L}_{finetune}(\theta) + \lambda_2 \|\theta - \theta_0\|_2^2$, and for $L_1$-regularization, it is $\mathcal{L}_{finetune}(\theta) + \lambda_1 \|\theta - \theta_0\|_1$, where $\lambda_2$ and $\lambda_1$ are the respective regularization hyperparameters.

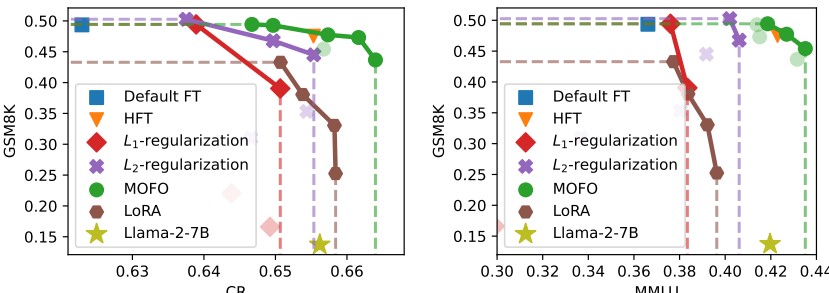

Figure 4: The performance on the math task (GSM8K) and the scores in general capabilities of Llama-2-7B after fine-tuning on the MetaMathQA dataset. Only points on the Pareto front are shown as solid points, while the remaining points are presented as semi-transparent. The results show that compared with $L_1$, $L_2$ regularization, and LoRA across various hyperparameter configurations, the MoFO algorithm achieves a better Pareto front.

We fine-tune the Llama-2-7B model on the MetaMathQA dataset using $L_1$ and $L_2$ regularization, as well as LoRA, and compare their performance with MoFO. We present the results in Figure 4 and plot Pareto optimal fronts[3] for these methods. Details of the hyperparameter configurations for this experiment are provided in Appendix D.3. These results show the effectiveness of the MoFO algorithm in both optimization and mitigating forgetting.

The result reveals that MoFO consistently achieves a better Pareto front in comparison to baseline methods. When compared to regularization methods and LoRA, MoFO exhibits less forgetting and can even maintain general capabilities with comparable GSM8K accuracies. Additionally, MoFO outperforms regularization methods in math tasks when the magnitudes of forgetting are similar. Additional experimental results using other LLMs and other datasets are provided in Appendix E.3.

### 4.3 CONTINUAL FINE-TUNING

In this section, we explore the performance of our proposed MoFO in continual fine-tuning on the TRACE benchmark (Wang et al., 2023b). We sequentially train TinyLlama-1.1B on the TRACE dataset, which includes the eight tasks from different domains. The implementation details are provided in Appendix D.

**Continual learning baselines.** We consider several traditional methods from the field of continual learning to compare with MoFO. These methods can also be orthogonal combined with MoFO to further enhance performance. **Replay** involves optimizing the model using current data along with a memory buffer containing samples from previous tasks to mitigate forgetting, and we follow the implementation in (Wang et al., 2023b). **Gradient of Episodic Memory (GEM)** (Lopez-Paz &

---

[3]Since it is impractical to exhaust all hyperparameter configurations in real experiments, we present linear interpolation approximations of the Pareto fronts in Figure 4.

Ranzato, 2017) mitigates forgetting by using gradients from old tasks to adjust the parameter updates during the training of new tasks. Elastic weight consolidation (EWC) (Kirkpatrick et al., 2017) uses Fisher information, approximated by gradients from previous tasks, to regularize parameter updates.

**Results of continual fine-tuning.** We present the experimental results of sequentially fine-tuning TinyLlama-1.1B on the TRACE benchmark with various methods in Table 2. The results indicate that in continual fine-tuning, MoFO not only outperforms other fine-tuning baselines but also surpasses GEM and EWC. Moreover, MoFO combines well with the Replay method, offering a 1.5% performance gain on the OP metric compared to using Replay alone. Moreover, MoFO also combines well with EWC offering at least a 2.1% performance gain on the OP metric compared to using EWC alone. Additionally, when combined with the GEM method, MoFO provides a 0.9% improvement on the OP metric compared to using GEM alone.

In summary, these results underscore the superior performance of MoFO in continual fine-tuning and its effectiveness in alleviating forgetting.

Table 2: The OP and BWT scores of TinyLlama-1.1B after fine-tuning on TRACE benchmark. The results show that MoFO outperforms Default FT, HFT, Proximal GD, GEM, and EWC in continual learning and can combine well with continual learning methods. Bold values denote the best results among these methods in each group.

|              | OP    | BWT   |
|--------------|-------|-------|
| Default FT   | 38.4  | -10.3 |
| HFT          | 39.9  | -10.1 |
| Proximal GD  | 38.2  | -11.2 |
| MoFO         | **41.3** | **-5.4** |
| GEM          | 40.8  | -8.5  |
| GEM + MoFO   | **41.7** | **-6.7** |
| EWC          | 41.1  | -8.3  |
| EWC + MoFO   | **43.2** | **-4.4** |
| Replay       | 45.5  | 4.7   |
| Replay + MoFO | **47.0** | **4.8** |

## 4.4 FURTHER ANALYSIS

**Impact of update strategy in MoFO.** In addition to MoFO, we consider three other BCD methods, **randomized BCD**, **gradient-filtered BCD**, and **MV-filtered BCD**. **Randomized BCD** updates a random subset of parameters at each iteration. **Gradient-filtered BCD** replaces MoFO's filter $\text{FLT}_\alpha(m_t)$ with $\text{FLT}_\alpha(g_t)$, while **MV-filtered BCD** uses $\text{FLT}_\alpha(m_t/\sqrt{v_t})$.

We fine-tune Llama-2-7B on MetaMathQA using these four methods with 10% parameter update fraction and present the results in Table 3. Experimental results show that all four BCD methods exhibit significantly less forgetting compared to Default FT, demonstrating the effectiveness of BCD algorithms in mitigating forgetting.

In terms of GSM8K performance, our proposed MoFO method significantly surpasses randomized BCD, Gradient-filtered BCD, and MV-filtered BCD, indicating that updating parameters with the largest momentum leads to strong optimization power.

Moreover, we provide analysis on the update fraction of parameters in MoFO in Appendix E.1. We also empirically verify that MoFO achieves its intended goal of converging to a minimum closer to the pre-trained model and reducing forgetting, as shown in Appendix E.2.

Table 3: The performance on the math reasoning task (GSM8K) and general capability scores of Llama-2-7B after fine-tuning on MetaMathQA using different updating strategies in MoFO.

| Method | GSM8K | General Capability | | | |
|--------|-------|------|------|-----------|------|
|        |       | CR   | MMLU | HumanEval | Avg. |
| Llama-2-7B | 13.7 | 65.6 | 42.0 | 24.2 | 43.9 |
| Default FT | 49.4 | 62.3 | 36.6 | 16.1 | 38.3 |
| *BCD methods ($\alpha\% = 10\%$)* | | | | | |
| Randomized BCD | 35.0 | 65.8 | 41.1 | 25.1 | 44.0 |
| Gradient-filtered BCD | 40.2 | **66.0** | 41.6 | **28.0** | 45.2 |
| MV-filtered BCD | 42.2 | **66.0** | 40.0 | 27.6 | 44.5 |
| MoFO | **45.4** | 65.7 | **43.5** | 27.4 | **45.5** |

## 5 WHY MoFO CONVERGES TO A CLOSER POINT? AN EXAMPLE

In this section, we conduct a preliminary analysis of the following question:

**Why does MoFO converge closer to the pre-trained LLMs than those of Adam?**

We attempt to answer this question by the following toy example. We denote $\theta = (\theta_1, \theta_2) \in \mathbb{R}^2$ to be the trainable parameters of our model and make the following assumptions: **the pre-training loss** is $\mathcal{L}_{pretrain}(\theta) = \theta_1^2 + \theta_2^2$ and the model has been trained to the global minimum $(0, 0)$ during the pre-trained phase; **the fine-tuning loss** is $\mathcal{L}_{finetune}(\theta) = (\theta_1 - 1)^2 (\theta_2 - 1)^2$. In this case, any global optimum of $\mathcal{L}_{finetune}$ lies in the set $\{(1, \theta_2) : \theta_2 \in \mathbb{R}\} \cup \{(\theta_1, 1) : \theta_1 \in \mathbb{R}\}$, which is a union of two straight lines.

For full-parameter fine-tuning with Adam, starting from $(0, 0)$, the model converges to $(1, 1)$ during the fine-tuning phase along the orange arrow in Figure 5, with a pre-training loss of 2. In contrast, when applying MoFO, the model converges to $(1, 0)$ during the fine-tuning phase along the green arrow in Figure 5, resulting in a pre-training loss of 1. This demonstrates that MoFO can converge to a minimum that is closer to the pre-training model, thereby mitigating forgetting.

**Intuition.** In this example, we find that when a loss function has multiple distinct minima, they can be considered as different attractors. These attractors can influence the gradient direction of a pre-trained model, possibly drawing the model's weights away from the nearest minimum. Specifically, full-parameter gradient descent based methods may converge to the balanced point of these attractors' influences, which is the orange point in Figure 5(a). On the contrary, MoFO addresses this issue by updating only a subset of parameters during each iteration. This selective updating rule reduces interference among attractors, allowing the model to converge to a closer minimum.

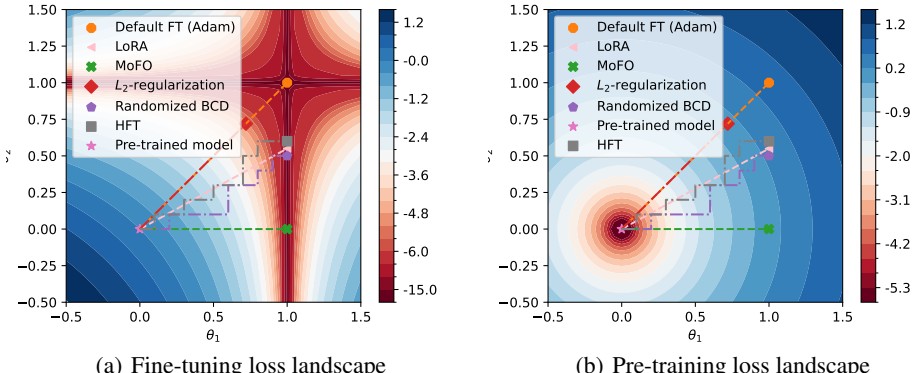

(a) Fine-tuning loss landscape      (b) Pre-training loss landscape

Figure 5: The loss landscapes of the example. We plot the landscapes on (a) the fine-tuning loss and (b) the pre-training loss. A logarithmic scale is applied to the loss values for better visualization. MoFO converges to a minimum closest to the pre-trained model, with a low pre-training loss.

## 6 CONCLUSION

This paper presents the Momentum-Filtered Optimizer (MoFO), a new approach designed to mitigate the crucial issue of pre-training knowledge forgetting in LLMs during fine-tuning. By selectively updating the parameters with the largest momentum magnitudes in each parameter block, MoFO converges to a point closer to the pre-trained model compared to full-parameter fine-tuning and effectively preserves pre-trained knowledge. Our experimental results demonstrate that MoFO not only achieves comparable performance to default fine-tuning but also effectively alleviates forgetting. Future work will explore further optimizations and potential applications of MoFO in RLHF.

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

# A SUPPLEMENTAL RELATED WORKS

**Block coordinate descent** Block Coordinate Descent (BCD) involves iteratively optimizing over a block of coordinates while holding the others constant. The foundational work of Tseng (2001) provides a comprehensive analysis of the convergence properties of BCD under certain conditions. Subsequent research has explored various BCD variants (Hong et al., 2017), including randomized BCD (Nesterov, 2012; Richtárik & Takáč, 2014; Lu & Xiao, 2015), cyclic BCD (Sun & Hong, 2015), and greedy BCD (Nutini et al., 2015). Among these, the greedy variant, also known as Gauss-Southwell BCD method, has drawn attention due to its ability to prioritize coordinates that yield the most substantial improvement in each iteration, thereby potentially accelerating convergence.

In the realm of machine learning, BCD has also found applications (Nutini et al., 2022). For example, Luo et al. (2024) leverages BCD to perform memory-efficient fine-tuning of LLM and Xu & Zhang (2024) uses random masking to perform this. In federated learning, Rothchild et al. (2020) adopts top-$k$ momentum value unsketch rather than our top-$k$ momentum filtering to tackle communication bottleneck and convergence issues. In LLMs, some concurrent works propose BCD-based algorithms leveraging task vectors to enhance fine-tuning performance (Li et al., 2024) and mitigate catastrophic forgetting in multi-task learning (Panda et al., 2024). In a recent work (Hui et al., 2024), catastrophic forgetting during the fine-tuning of LLMs is addressed by selectively freezing 50% of the model parameters during training. Our approach is akin to a more efficient greedy BCD, achieving superior performance in fine-tuning tasks and alleviating forgetting better.

## B Supplementary Analysis on the Top-$\alpha\%$ Filter

In this section, we provide supplementary analysis on our top-$\alpha\%$ filter, which serves as a preliminary for proving Theorem 1 in Appendix C.

As introduced in Section D.4, the entire parameter space is divided into $B$ parts, with the $k$-th part having a dimension of $d_k$. We assume the parameter space is $\mathbb{R}^d$, which can be expressed as the product $\mathbb{R}^d \cong \mathbb{R}^{d_1} \times \mathbb{R}^{d_2} \times \cdots \times \mathbb{R}^{d_B}$. For any $z \in \mathbb{R}^d$, we represent it as:

$$z = \text{Concat}(z^{(1)}, z^{(2)}, \ldots, z^{(B)}),$$

where $z^{(k)} \in \mathbb{R}^{d_k}$ for each $1 \leq k \leq B$.

**Definition 1.** *For any $z \in \mathbb{R}^d$, we define the top-$\alpha\%$ filter of $z$ as*

$$FLT_\alpha(z) := \text{Concat}(\mathbf{e}_{S_1}^{(1)}; \mathbf{e}_{S_2}^{(2)}; \ldots; \mathbf{e}_{S_B}^{(B)}) \in \mathbb{R}^d,$$

*where*

$$S_k = \left\{ i \in [d_k] : |z_i^{(k)}| \text{ ranks within the top-}\alpha\% \text{ of all } |z^{(k)}|\text{'s entries } (|z_1^{(k)}|, |z_2^{(k)}|, \ldots, |z_{d_k}^{(k)}|) \right\}$$

*and $\mathbf{e}_{S_k}^{(k)}$ is a $d_k$-dimensional vector where the $i$-th entry is 1 if $i \in S_k$, and 0 otherwise.*

**Remark 1.** *To ensure that the top-$\alpha\%$ filter $FLT_\alpha(z)$ is well-defined, when multiple entries share identical absolute values and including all of them in the set $S_k$ would result in exceeding the $\alpha\%$ threshold of set size, the construction of $S_k$ prioritizes the entries with the smallest indices among those with the same absolute values.*

**Definition 2.** *For any $z \in \mathbb{R}^d$, we define the $L_{1,\text{top-}\alpha\%}$ norm of $z$ as*

$$\|z\|_{1,\text{top-}\alpha\%} := \|z \odot FLT_\alpha(z)\|_1.$$

**Proposition 1.** $\|\cdot\|_{1,\text{top-}\alpha\%}$ *is indeed a norm in $\mathbb{R}^d$.*

*Proof.* By Definition 1, we get

$$\|z\|_{1,\text{top-}\alpha\%} = \|z \odot FLT_\alpha(z)\|_1 = \sum_{k=1}^{B} \|z^{(k)} \odot \mathbf{e}_{S_k}^{(k)}\|_1. \tag{5}$$

First, if $\|z\|_{1,\text{top-}\alpha\%} = 0$, then by (5), $\|z^{(k)} \odot \mathbf{e}_{S_k}^{(k)}\|_1 = 0$ for any $1 \leq k \leq B$. Thus,

$$\|z^{(k)}\|_\infty = \underset{1 \leq i \leq d_k}{\arg\max} |z_i^{(k)}| \leq \|z^{(k)} \odot \mathbf{e}_{S_k}^{(k)}\|_1 = 0.$$

So $z^{(k)}$ is a zero vector for any $1 \leq k \leq B$ and then $z$ is a zero vector.

Second, for any given $c \in \mathbb{R}_+$, $\{|z_i^{(k)}|\}_{1 \leq i \leq d_k}$ and $\{|cz_i^{(k)}|\}_{1 \leq i \leq d_k}$ have the same order. So $z$ and $cz$ share the same filter $FLT_\alpha(z)$ and

$$\|cz\|_{1,\text{top-}\alpha\%} = \|cz \odot FLT_\alpha(cz)\|_1 = c\|z \odot FLT_\alpha(z)\|_1 = c\|z\|_{1,\text{top-}\alpha\%}.$$

Third, for any $x, y \in \mathbb{R}^d$, suppose that

$$FLT_\alpha(x) = \text{Concat}(\mathbf{e}_{S_1'}^{(1)}; \mathbf{e}_{S_2'}^{(2)}; \ldots; \mathbf{e}_{S_B'}^{(B)}) \quad \text{and} \quad FLT_\alpha(x+y) = \text{Concat}(\mathbf{e}_{S_1''}^{(1)}; \mathbf{e}_{S_2''}^{(2)}; \ldots; \mathbf{e}_{S_B''}^{(B)}).$$

By the construction of $S_k'$, for any $1 \leq k \leq B$, we have

$$\|x^{(k)} \odot \mathbf{e}_{S_k''}^{(k)}\|_1 \leq \|x^{(k)} \odot \mathbf{e}_{S_k'}^{(k)}\|_1.$$

So

$$\|x \odot FLT_\alpha(x+y)\|_1 = \sum_{k=1}^{B} \|x^{(k)} \odot \mathbf{e}_{S_k''}^{(k)} \leq \sum_{k=1}^{B} \|x^{(k)} \odot \mathbf{e}_{S_k'}^{(k)} = \|x \odot FLT_\alpha(x)\|_1.$$

Similarly, it holds that

$$\|y \odot FLT_\alpha(x+y)\|_1 \leq \|y \odot FLT_\alpha(y)\|_1.$$

Thus, we have

$$\begin{aligned}
\|x+y\|_{1,\text{top-}\alpha\%} &= \|(x+y) \odot \text{FLT}_\alpha(x+y)\|_1 \\
&= \|x \odot \text{FLT}_\alpha(x+y) + y \odot \text{FLT}_\alpha(x+y)\|_1 \\
&\leq \|x \odot \text{FLT}_\alpha(x+y)\|_1 + \|y \odot \text{FLT}_\alpha(x+y)\|_1 \\
&\leq \|x \odot \text{FLT}_\alpha(x)\|_1 + \|y \odot \text{FLT}_\alpha(y)\|_1 \\
&= \|x\|_{1,\text{top-}\alpha\%} + \|y\|_{1,\text{top-}\alpha\%}.
\end{aligned}$$

$\square$

We propose a lemma which is useful for the proof of Theorem 1.

**Lemma 1.** *For any $x, y \in \mathbb{R}^d$, it holds that*

$$\|x \odot FLT_\alpha(x)\|_1 - \|x \odot FLT_\alpha(y)\|_1 \leq 2\|x - y\|_1.$$

*Proof.* By Proposition 1, $\|\cdot\|_{1,\text{top-}\alpha\%}$ is a norm in $\mathbb{R}^d$, so we have

$$\begin{aligned}
&\|x \odot \text{FLT}_\alpha(x)\|_1 - \|x \odot \text{FLT}_\alpha(y)\|_1 \\
&= \|x \odot \text{FLT}_\alpha(x)\|_1 - \|y \odot \text{FLT}_\alpha(y)\|_1 + \|y \odot \text{FLT}_\alpha(y)\|_1 - \|x \odot \text{FLT}_\alpha(y)\|_1 \\
&= \|x\|_{1,\text{top-}\alpha\%} - \|y\|_{1,\text{top-}\alpha\%} + \|y \odot \text{FLT}_\alpha(y)\|_1 - \|x \odot \text{FLT}_\alpha(y)\|_1 \\
&\leq \|x - y\|_{1,\text{top-}\alpha\%} + \|(y - x) \odot \text{FLT}_\alpha(y)\|_1 \\
&\leq \|x - y\|_1 + \|y - x\|_1 \\
&= 2\|x - y\|_1.
\end{aligned}$$

$\square$

## C    PROOF OF THEOREM 1

Our proof of Theorem 1 follows the convergence analysis of the full-batch Adam optimizer in Shi et al. (2021), with novel adaptations to address the unique aspects of MoFO.

To maintain consistency with the notation used in MoFO (Algorithm 1 in Section D.4), we denote

$$z_t = \texttt{Concat}(z_t^{(1)}, \ldots, z_t^{(B)}),$$

where $z$ represents the model parameter $\theta$, the gradient $g$, the first moment estimate $m$, or the second moment estimate $v$. Notably, each of these variables belongs to $\mathbb{R}^d$. Thus, for any $1 \leq i \leq d$, we can denote $z_{i,t}$ as the $i$-th entry of $z_t$ when $z$ represents $\theta$, $g$, $m$, or $v$.

By the update rules of the first and second moment estimates

$$m_{i,t} = (1 - \beta_1)g_{i,t} + \beta_1 m_{i,t-1}, \quad m_{i,0} = 0,$$

$$v_{i,t} = (1 - \beta_2)g_{i,t}^2 + \beta_2 v_{i,t-1}, \quad v_{i,0} = 0.$$

By mathematical induction, for any $1 \leq i \leq d$, we have

$$m_{i,t} = (1 - \beta_1)\sum_{s=1}^{t} \beta_1^{t-s} g_{i,s} \tag{6}$$

and

$$v_{i,t} = (1 - \beta_2)\sum_{s=1}^{t} \beta_2^{t-s} g_{i,s}^2. \tag{7}$$

We will frequently use Equation (6) and (7) in the proofs of the subsequent lemmas and theorems.

**Lemma 2.** *For the full-batch version of MoFO with hyperparameters satisfying $\beta_1 < \sqrt{\beta_2} < 1$, $\epsilon = 0$, it holds that*

$$|\theta_{i,t} - \theta_{i,t-1}| \leq \frac{1}{\sqrt{1 - \beta_2}(1 - \beta_1/\sqrt{\beta_2})} \cdot \eta_t \cdot \texttt{FLT}_\alpha(m_t)_i, \quad \text{for any coordinate } 1 \leq i \leq d.$$

*Moreover, it holds that*

$$\|\theta_t - \theta_{t-1}\|_2 \leq C\eta_t,$$

*where $C = \frac{\sqrt{d \cdot (\alpha\%) + B}}{\sqrt{1 - \beta_2}(1 - \beta_1/\sqrt{\beta_2})}$.*

*Proof.* When the $i$-th entry is not in our filter at iteration $t$, i.e. $\texttt{FLT}_\alpha(m_t)_i = 0$, we have $\theta_{i,t} = \theta_{i,t-1}$. Then

$$|\theta_{i,t} - \theta_{i,t-1}| = 0 = \frac{1}{\sqrt{1 - \beta_2}(1 - \beta_1/\sqrt{\beta_2})} \cdot \eta_t \cdot \texttt{FLT}_\alpha(m_t)_i.$$

When the $i$-th entry is in our filter, i.e. $\texttt{FLT}_\alpha(m_t)_i = 1$, by the weight updating rule of MoFO, we have $\theta_{i,t} - \theta_{i,t-1} = -\eta_t \hat{m}_{i,t}/\sqrt{\hat{v}_{i,t}}$. We first analyze $m_{i,t}$ and $v_{i,t}$.

By Equation (6) and (7), we get

$$|m_{i,t}| \leq (1 - \beta_1)\sum_{s=1}^{t} \beta_1^{t-s} |g_{i,s}|,$$

$$v_{i,t} = (1 - \beta_2)\sum_{s=1}^{t} \beta_2^{t-s} g_{i,s}^2 \geq (1 - \beta_2)\beta_2^{t-s} g_{i,s}^2, \quad \text{for any } 1 \leq s \leq t.$$

So we get

$$
\left|\theta_{i,t} - \theta_{i,t-1}\right| = \left|-\eta_t \frac{\hat{m}_{i,t}}{\sqrt{\hat{v}_{i,t}}}\right| = \eta_t \frac{\sqrt{1-\beta_2^t}}{1-\beta_1^t} |m_{i,t}|/\sqrt{v_{i,t}}
$$

$$
\leq \eta_t \frac{\sqrt{1-\beta_2^t}}{1-\beta_1^t} \sum_{s=1}^{t} \frac{(1-\beta_1)\beta_1^{t-s}|g_{i,s}|}{\sqrt{(1-\beta_2)\beta_2^{t-s}|g_{i,s}|}} = \eta_t \frac{1-\beta_1}{1-\beta_1^t} \sqrt{\frac{1-\beta_2^t}{1-\beta_2}} \sum_{s=1}^{t} (\beta_1/\sqrt{\beta_2})^{t-s}
$$

$$
\leq \frac{\eta_t}{\sqrt{1-\beta_2}} \sum_{s=0}^{t-1} (\beta_1/\sqrt{\beta_2})^s
$$

$$
\leq \frac{\eta_t}{\sqrt{1-\beta_2}(1-\beta_1/\sqrt{\beta_2})}.
$$

Here, the last inequality holds because of the assumption $\beta_1 < \sqrt{\beta_2} < 1$.

MoFO actually choose $\lceil d_k \times \alpha\% \rceil$ entries to update in each part $k$ of parameters. Then for any $z \in \mathbb{R}^d$, we have

$$
\#\{1 \leq i \leq d : \text{FLT}_\alpha(z)_i = 1\} = \sum_{k=1}^{B} \lceil d_k \cdot (\alpha\%) \rceil \leq \sum_{k=1}^{B} (d_k \cdot (\alpha\%) + 1) = d \cdot (\alpha\%) + B.
$$

Then for the $L_2$-distance, we have

$$
\|\theta_t - \theta_{t-1}\|_2 = \left( \sum_{k=1}^{d} |\theta_{i,t} - \theta_{i,t-1}|^2 \cdot \text{FLT}_\alpha(m_t)_i \right)^{\frac{1}{2}}
$$

$$
\leq \left( \frac{\eta_t^2}{(\sqrt{1-\beta_2}(1-\beta_1/\sqrt{\beta_2}))^2} \cdot \#\{1 \leq i \leq d : \text{FLT}_\alpha(z)_i = 1\} \right)^{\frac{1}{2}}
$$

$$
\leq \frac{\sqrt{d \cdot (\alpha\%) + B}}{\sqrt{1-\beta_2}(1-\beta_1/\sqrt{\beta_2})} \cdot \eta_t
$$

$$
= C\eta_t.
$$

$\square$

**Lemma 3.** *Suppose that the gradient $\nabla\mathcal{L}$ is Lipschitz continuous with constant $L$. Suppose that the full-batch version of MoFO has the hyperparameters satisfying $\beta_1 < \sqrt{\beta_2} < 1$, $\epsilon = 0$ and the learning rate schedule $\eta_t = \eta/\sqrt{t}$. For any iteration steps $t \geq s \geq 1$ and any coordinate $i$, it holds that*

$$
|g_{i,t} - g_{i,s}| \leq \|g_t - g_s\|_2 \leq \frac{2\sqrt{2}LC\eta(t-s)}{\sqrt{t}}.
$$

*Proof.* Since $\nabla\mathcal{L}$ has Lipschitz constant $L$, we get

$$
|g_{i,t} - g_{i,s}| \leq \|g_t - g_s\|_2 = \|\nabla\mathcal{L}(\theta_{t-1}) - \nabla\mathcal{L}(\theta_{t-1})\|_2 \leq L\|\theta_{t-1} - \theta_{s-1}\|_2. \tag{8}
$$

By Lemma 2, for any $t > s \geq 1$, we have

$$\|\theta_{t-1} - \theta_{s-1}\|_2 \leq \sum_{u=s}^{t-1} \|\theta_u - \theta_{u-1}\|_2 \leq C \sum_{u=s}^{t-1} \eta_u$$

$$\leq C\eta \sum_{u=s}^{t-1} \frac{1}{\sqrt{u}} \leq C\eta \sum_{u=s}^{t-1} \frac{2}{\sqrt{u-1}+\sqrt{u}} \leq 2C\eta \sum_{u=s}^{t-1} (\sqrt{u} - \sqrt{u-1})$$

$$= 2C\eta(\sqrt{t-1} - \sqrt{s-1}) = \frac{2C\eta(t-s)}{\sqrt{t-1}+\sqrt{s-1}}$$

$$\leq \frac{2C\eta(t-s)}{\sqrt{t-1}} \leq \frac{2C\eta(t-s)}{\sqrt{t/2}}$$

$$= \frac{2\sqrt{2}C\eta(t-s)}{\sqrt{t}}.$$

When $t = s > 1$, it is obvious that

$$\|\theta_{t-1} - \theta_{s-1}\|_2 = 0 \leq \frac{2\sqrt{2}C\eta(t-s)}{\sqrt{t}}.$$

Combining it with (8), for any $t \geq s \geq 1$, we have

$$|g_{i,t} - g_{i,s}| \leq \|g_t - g_s\|_2 \leq \frac{2\sqrt{2}LC\eta(t-s)}{\sqrt{t}}.$$

$\square$

**Lemma 4.** *Under the assumptions in Lemma 3, for any iteration step $t \geq 1$ and any coordinate $i$, it holds that*

$$g_{i,t}\frac{\hat{m}_{i,t}}{\sqrt{\hat{v}_{i,t}}} \geq \sqrt{1-\beta_2}\left(|g_{i,t}| - \left[\frac{2\sqrt{2}\beta_1}{(1-\beta_1)^2} + \frac{4}{1-\beta_2}\right]\frac{LC\eta}{\sqrt{t}}\right).$$

*Proof.* By Lemma 3, we get

$$g_{i,t}g_{i,s} = g_{i,t}^2 - g_{i,t}(g_{i,t} - g_{i,s}) \geq g_{i,t}^2 - |g_{i,t}| \cdot |g_{i,t} - g_{i,s}| \geq g_{i,t}^2 - \frac{2\sqrt{2}LC\eta(t-s)}{\sqrt{t}}|g_{i,t}|.$$

Then we have

$$g_{i,t}m_{i,t} = (1-\beta_1)\sum_{s=1}^{t}\beta_1^{t-s}g_{i,t}g_{i,s}$$

$$\geq g_{i,t}^2 \cdot (1-\beta_1)\sum_{s=1}^{t}\beta_1^{t-s} - \frac{2\sqrt{2}LC\eta}{\sqrt{t}}|g_{i,t}| \cdot (1-\beta_1)\sum_{s=1}^{t}\beta_1^{t-s} \cdot (t-s) \qquad (9)$$

$$\geq g_{i,t}^2 \cdot (1-\beta_1)\sum_{s=0}^{t-1}\beta_1^{s} - \frac{2\sqrt{2}LC\eta}{\sqrt{t}}|g_{i,t}| \cdot (1-\beta_1)\sum_{s=1}^{t-1}s\beta_1^{s}.$$

Since we have

$$\sum_{s=0}^{t-1}\beta_1^{s} = \frac{1-\beta_1^t}{1-\beta_1}, \quad \sum_{s=1}^{t-1}s\beta_1^{s-1} \leq \sum_{s=1}^{\infty}s\beta_1^{s-1} = \frac{d}{d\beta_1}\left(\sum_{s=1}^{\infty}\beta_1^{s}\right) = \frac{d}{d\beta_1}\left(\frac{\beta_1}{1-\beta_1}\right) = \frac{1}{(1-\beta_1)^2}, \qquad (10)$$

it holds that

$$g_{i,t}m_{i,t} \geq \text{RHS of } (9) \geq (1-\beta_1^t)g_{i,t}^2 - \frac{2\sqrt{2}\beta_1 LC\eta}{(1-\beta_1)\sqrt{t}}|g_{i,t}|. \qquad (11)$$

For the second moment estimate, we have

$$v_{i,t} = (1 - \beta_2) \sum_{s=1}^{t} \beta_2^{t-s} g_{i,s}^2 \leq (1 - \beta_2) \sum_{s=1}^{t} \beta_2^{t-s} (|g_{i,t}| + |g_{i,s} - g_{i,t}|)^2$$

$$\leq (1 - \beta_2) \sum_{s=1}^{t} \beta_2^{t-s} \left( |g_{i,t}| + \frac{2\sqrt{2}LC\eta(t-s)}{\sqrt{t}} \right)^2 = (1 - \beta_2) \sum_{s=0}^{t-1} \beta_2^s \left( |g_{i,t}| + \frac{2\sqrt{2}LC\eta s}{\sqrt{t}} \right)^2$$

$$= |g_{i,t}|^2 \cdot (1 - \beta_2) \left( \sum_{s=0}^{t-1} \beta_2^s \right) + |g_{i,t}| \cdot \frac{4\sqrt{2}LC\eta}{\sqrt{t}} (1 - \beta_2) \left( \sum_{s=1}^{t-1} s\beta_2^s \right)$$

$$+ \frac{8L^2C^2\eta^2}{t} (1 - \beta_2) \left( \sum_{s=1}^{t-1} s^2 \beta_2^s \right). \tag{12}$$

Since we have

$$\sum_{s=0}^{t-1} \beta_2^s = \frac{1 - \beta_2^t}{1 - \beta_2} \leq \frac{1}{1 - \beta_2},$$

$$\sum_{s=0}^{t-1} s\beta_2^{s-1} \leq \sum_{s=0}^{\infty} s\beta_2^{s-1} = \frac{d}{d\beta_2} \left( \sum_{s=0}^{\infty} \beta_2^s \right) = \frac{d}{d\beta_2} \left( \frac{1}{1 - \beta_2} \right) = \frac{1}{(1 - \beta_2)^2},$$

$$\sum_{s=0}^{t-1} s^2 \beta_2^{s-1} \leq \sum_{s=0}^{\infty} s^2 \beta_2^{s-1} = \beta_2 \left( \sum_{s=0}^{\infty} s(s-1)\beta_2^{s-2} \right) + \sum_{s=0}^{\infty} s\beta_2^{s-1}$$

$$= \beta_2 \cdot \frac{d^2}{d\beta_2^2} \left( \sum_{s=0}^{\infty} \beta_2^s \right) + \frac{1}{(1 - \beta_2)^2} = \beta_2 \cdot \frac{d^2}{d\beta_2^2} \left( \frac{1}{1 - \beta_2} \right) + \frac{1}{(1 - \beta_2)^2}$$

$$= \frac{2\beta_2}{(1 - \beta_2)^3} + \frac{1}{(1 - \beta_2)^2}$$

$$= \frac{1 + \beta_2}{(1 - \beta_2)^3},$$

it holds that

$$v_{i,t} \leq \text{RHS of (12)} \leq |g_{i,t}|^2 + |g_{i,t}| \cdot \frac{4\sqrt{2}\beta_2 LC\eta}{(1 - \beta_2)\sqrt{t}} + \frac{8(1 + \beta_2)\beta_2 L^2 C^2 \eta^2}{(1 - \beta_2)^2 t}$$

$$\leq |g_{i,t}|^2 + |g_{i,t}| \cdot \frac{8LC\eta}{(1 - \beta_2)\sqrt{t}} + \frac{16L^2C^2\eta^2}{(1 - \beta_2)^2 t}$$

$$= \left( |g_{i,t}| + \frac{4LC\eta}{(1 - \beta_2)\sqrt{t}} \right)^2.$$

Thus, we get

$$\sqrt{v_{i,t}} \leq |g_{i,t}| + \frac{4LC\eta}{(1 - \beta_2)\sqrt{t}}.$$

Recalling (11), we have

$$
\begin{aligned}
g_{i,t} m_{i,t} &\geq (1 - \beta_1^t)\left(|g_{i,t}| + \frac{4LC\eta}{(1-\beta_2)\sqrt{t}}\right)\left(|g_{i,t}| - \frac{2\sqrt{2}\beta_1 LC\eta}{(1-\beta_1^t)(1-\beta_1)\sqrt{t}} - \frac{4LC\eta}{(1-\beta_2)\sqrt{t}}\right) \\
&\quad + (1 - \beta_1^t) \cdot \frac{4LC\eta}{(1-\beta_2)\sqrt{t}}\left(\frac{2\sqrt{2}\beta_1 LC\eta}{(1-\beta_1^t)(1-\beta_1)\sqrt{t}} + \frac{4LC\eta}{(1-\beta_2)\sqrt{t}}\right) \\
&\geq (1 - \beta_1^t)\left(|g_{i,t}| + \frac{4LC\eta}{(1-\beta_2)\sqrt{t}}\right)\left(|g_{i,t}| - \frac{2\sqrt{2}\beta_1 LC\eta}{(1-\beta_1^t)(1-\beta_1)\sqrt{t}} - \frac{4LC\eta}{(1-\beta_2)\sqrt{t}}\right) \\
&\geq (1 - \beta_1^t)\sqrt{v_{i,t}}\left(|g_{i,t}| - \frac{2\sqrt{2}\beta_1 LC\eta}{(1-\beta_1^t)(1-\beta_1)\sqrt{t}} - \frac{4LC\eta}{(1-\beta_2)\sqrt{t}}\right).
\end{aligned}
$$

Therefore,

$$
\begin{aligned}
g_{i,t}\frac{\hat{m}_{i,t}}{\sqrt{\hat{v}_{i,t}}} = \frac{\sqrt{1-\beta_2^t}}{1-\beta_1^t}g_{i,t}\frac{m_{i,t}}{\sqrt{v_{i,t}}} &\geq \sqrt{1-\beta_2^t}\left(|g_{i,t}| - \frac{2\sqrt{2}\beta_1 LC\eta}{(1-\beta_1^t)(1-\beta_1)\sqrt{t}} - \frac{4LC\eta}{(1-\beta_2)\sqrt{t}}\right) \\
&\geq \sqrt{1-\beta_2}\left(|g_{i,t}| - \left[\frac{2\sqrt{2}\beta_1}{(1-\beta_1)^2} + \frac{4}{1-\beta_2}\right]\frac{LC\eta}{\sqrt{t}}\right).
\end{aligned}
$$

$\square$

**Lemma 5.** *Under the assumptions in Lemma 3, for any iteration step $t \geq 1$ and any coordinate $i$, it holds that*

$$
\left\|\frac{m_t}{1-\beta_1^t} - g_t\right\|_1 \leq \frac{2\sqrt{2}\beta_1\sqrt{d}LC\eta}{(1-\beta_1)^2\sqrt{t}}.
$$

*Proof.* Recalling (6), we get

$$
m_t = (1 - \beta_1)\sum_{s=1}^{t}\beta_1^{t-s}g_s,
$$

and

$$
m_t - (1-\beta_1^t)g_t = (1-\beta_1)\sum_{s=1}^{t}\beta_1^{t-s}(g_t - g_s).
$$

By Lemma 3 and Equation (10) in the proof of Lemma 4, we get

$$
\begin{aligned}
\left\|\frac{m_t}{1-\beta_1^t} - g_t\right\|_2 &\leq \frac{1-\beta_1}{1-\beta_1^t}\sum_{s=1}^{t}\beta_1^{t-s}\|g_t - g_s\|_2 \leq \sum_{s=1}^{t}\beta_1^{t-s}\|g_t - g_s\|_2 \\
&\leq \frac{2\sqrt{2}LC\eta}{\sqrt{t}}\sum_{s=1}^{t}\beta_1^{t-s}(t-s) = \frac{2\sqrt{2}LC\eta}{\sqrt{t}}\sum_{s=0}^{t-1}s\beta_1^s \\
&\leq \frac{2\sqrt{2}\beta_1 LC\eta}{(1-\beta_1)^2\sqrt{t}}.
\end{aligned}
$$

By Cauchy-Schwarz's inequality, we have

$$
\left\|\frac{m_t}{1-\beta_1^t} - g_t\right\|_1 \leq \sqrt{d}\left\|\frac{m_t}{1-\beta_1^t} - g_t\right\|_2 \leq \frac{2\sqrt{2}\beta_1\sqrt{d}LC\eta}{(1-\beta_1)^2\sqrt{t}}.
$$

$\square$

Now we will complete the proof of Theorem 1.

*Proof of Theorem 1.* By the descent lemma, since $\nabla\mathcal{L}$ is Lipschitz with constant $L$, we have

$$\mathcal{L}(\theta_t) - \mathcal{L}(\theta_{t-1}) \leq \nabla\mathcal{L}(\theta_{t-1})^\top(\theta_t - \theta_{t-1}) + \frac{L}{2}\|\theta_t - \theta_{t-1}\|_2^2$$

$$\leq g_t^\top(\theta_t - \theta_{t-1}) + \frac{L}{2}\|\theta_t - \theta_{t-1}\|_2^2. \tag{13}$$

By Lemma 2 and Lemma 4, we have

$$\mathcal{L}(\theta_t) - \mathcal{L}(\theta_{t-1}) \leq \text{RHS of } (13) \leq -\eta_t\left(\sum_{i=1}^d g_{i,t}\frac{\hat{m}_{i,t}}{\sqrt{\hat{v}_{i,t}}}\cdot\texttt{FLT}_\alpha(m_t)_i\right) + \frac{LC^2\eta_t^2}{2}$$

$$\leq \frac{LC^2\eta^2}{2t} - \frac{\eta}{\sqrt{t}}\sum_{i=1}^d\sqrt{1-\beta_2}\left(|g_{i,t}| - \left[\frac{2\sqrt{2}\beta_1}{(1-\beta_1)^2} + \frac{4}{1-\beta_2}\right]\frac{LC\eta}{\sqrt{t}}\right)\cdot\texttt{FLT}_\alpha(m_t)_i$$

$$= -\frac{\sqrt{1-\beta_2}\cdot\eta}{\sqrt{t}}\|g_t\odot\texttt{FLT}_\alpha(m_t)\|_1 + \left[\frac{2\sqrt{2}\beta_1\sqrt{1-\beta_2}}{(1-\beta_1)^2} + \frac{4}{\sqrt{1-\beta_2}} + \frac{C}{2}\right]\frac{LC\eta^2}{t}\cdot\|\texttt{FLT}_\alpha(m_t)\|_1$$

$$\leq -\frac{\sqrt{1-\beta_2}\cdot\eta}{\sqrt{t}}\|g_t\odot\texttt{FLT}_\alpha(m_t)\|_1 + \left[\frac{2\sqrt{2}\beta_1\sqrt{1-\beta_2}}{(1-\beta_1)^2} + \frac{4}{\sqrt{1-\beta_2}} + \frac{C}{2}\right]\frac{LC\eta^2(d\cdot(\alpha\%)+B)}{t}.$$

$$\tag{14}$$

By Lemma 1 and Lemma 5, we have

$$\|g_t\odot\texttt{FLT}_\alpha(g_t)\|_1 - \|g_t\odot\texttt{FLT}_\alpha(m_t)\|_1 = \|g_t\odot\texttt{FLT}_\alpha(g_t)\|_1 - \left\|g_t\odot\texttt{FLT}_\alpha\left(\frac{m_t}{1-\beta_1^t}\right)\right\|_1$$

$$\leq 2\left\|g_t - \frac{m_t}{1-\beta_1^t}\right\|_1$$

$$\leq \frac{4\sqrt{2}\beta_1\sqrt{d}LC\eta}{(1-\beta_2)^2\sqrt{t}}.$$

Thus,

$$\mathcal{L}(\theta_t) - \mathcal{L}(\theta_{t-1}) \leq \text{RHS of } (14)$$

$$\leq -\frac{\sqrt{1-\beta_2}\cdot\eta}{\sqrt{t}}\|g_t\odot\texttt{FLT}_\alpha(g_t)\|_1 + \left[\frac{2\sqrt{2}\beta_1\sqrt{1-\beta_2}}{(1-\beta_1)^2} + \frac{4}{\sqrt{1-\beta_2}} + \frac{C}{2}\right]\frac{LC\eta^2(d\cdot(\alpha\%)+B)}{t}$$

$$+ \frac{4\sqrt{2}\beta_1\sqrt{d}LC\eta^2}{(1-\beta_2)^{\frac{3}{2}}t}$$

$$= -\frac{C_1}{\sqrt{t}}\|g_t\|_{1,\text{top-}\alpha\%} + \frac{C_2}{t} \leq -\frac{C_1}{\sqrt{t}}\min_{1\leq t\leq T}\|g_t\|_{1,\text{top-}\alpha\%} + \frac{C_2}{t},$$

$$\tag{15}$$

where

$$C_1 = \sqrt{1-\beta_2}\cdot\eta,$$

$$C_2 = LC\eta^2\cdot\left\{\left[\frac{2\sqrt{2}\beta_1\sqrt{1-\beta_2}}{(1-\beta_1)^2} + \frac{4}{\sqrt{1-\beta_2}} + \frac{C}{2}\right](d\cdot(\alpha\%)+B) + \frac{4\sqrt{2}\beta_1\sqrt{d}}{(1-\beta_2)^{\frac{3}{2}}}\right\}.$$

Taking the summation of (14) from 1 to $T$, we get

$$\mathcal{L}^* - \mathcal{L}(\theta_0) \leq \mathcal{L}(\theta_T) - \mathcal{L}(\theta_0) = \sum_{t=1}^T\mathcal{L}(\theta_t) - \mathcal{L}(\theta_{t-1})$$

$$\leq -C_1\left(\sum_{t=1}^T\frac{1}{\sqrt{t}}\right)\cdot\min_{1\leq t\leq T}\|g_t\odot\texttt{FLT}_\alpha(g_t)\|_1 + C_2\sum_{t=1}^T\frac{1}{t}.$$

Since

$$\sum_{t=1}^{T} \frac{1}{\sqrt{t}} \geq \sum_{t=1}^{T} \frac{2}{\sqrt{t}+\sqrt{t+1}} = \sum_{t=1}^{T} 2(\sqrt{t+1}-\sqrt{t}) = 2(\sqrt{T+1}-1),$$

$$\sum_{t=1}^{T} \frac{1}{t} = 1 + \sum_{t=1}^{T-1} \frac{1}{t+1} \leq 1 + \sum_{t=1}^{T-1} \int_{t}^{t+1} \frac{1}{u}\,du \leq 1 + \int_{1}^{T} \frac{1}{u}\,du = 1 + \log T,$$

we get

$$\min_{0 \leq t \leq T-1} \|\nabla\mathcal{L}(\theta_t)\|_\infty = \min_{1 \leq t \leq T} \|g_t\|_\infty \leq \min_{1 \leq t \leq T} \|g_t\|_{1,\text{top-}\alpha\%}$$

$$\leq \frac{\mathcal{L}(\theta_0) - \mathcal{L}^* + C_2 \sum_{t=1}^{T} \frac{1}{t}}{C_1 \sum_{t=1}^{T} \frac{1}{\sqrt{t}}} \leq \frac{\mathcal{L}(\theta_0) - \mathcal{L}^* + C_2(1 + \log T)}{2C_1(\sqrt{T+1}-1)}$$

$$= \mathcal{O}\left(\frac{\log T}{\sqrt{T}}\right).$$

$\square$

# D IMPLEMENTATION DETAILS

## D.1 DATASETS FOR FINE-TUNING.

**MetaMathQA** (Yu et al., 2024b). This dataset comprises 395K math question-answer pairs. Numerous studies indicate that LLMs significantly enhance performance metrics on mathematical benchmarks such as GSM8K after fine-tuning on this dataset. We randomly select 10% of this dataset for training LLMs, which includes 39.5K question-answer pairs.

**PMC-LLaMA-Instructions** (Wu et al., 2024). This dataset comprises 514K instruction-response pairs. Fine-tuning LLMs on this dataset has been shown to enhance performance on medical NLP tasks, such as PubMedQA (Jin et al., 2019), MedMCQA (Pal et al., 2022), and MedQA (Jin et al., 2021). We randomly sampled 51K instances with prompt lengths less than 750 characters for training our models.

**TRACE benchmark dataset** (Wang et al., 2023b). TRACE benchmark is designed with a comprehensive set of 8 distinct tasks across various domains, including domain-specific knowledge, multilingual proficiency, code generation, and mathematical reasoning.

## D.2 EVALUATION METRICS FOR INSTRUCTION FINE-TUNING

We employ a comprehensive suite of widely used benchmarks to assess the performance and potential catastrophic forgetting effects on the general capabilities of LLMs after instruction fine-tuning. The benchmarks are as follows:

- **Factual knowledge (MMLU)**: We use the Massive Multitask Language Understanding (MMLU) benchmark (Hendrycks et al., 2021) to evaluate factual knowledge across 57 diverse subjects, ranging from STEM fields and the humanities to social sciences. Evaluations are performed using 8-bit precision with the open-instruct implementation, and by following the setup of (Hui et al., 2024), we report the 0-shot accuracy.

- **Common sense reasoning (CommonSense)**: To measure the commonsense reasoning capabilities of LLMs, we employ the widely recognized benchmarks ARC-Challenge, ARC-Easy (Clark et al., 2018), and HellaSwag (Zellers et al., 2019), collectively referred to as the Commonsense benchmark. We use the average of their metrics as the evaluation, conducting assessments using the LM Eval Harness framework (Gao et al., 2023) and reporting the 0-shot accuracy based on the "acc_norm, none" metric.

- **Mathematical Reasoning (GSM8K)**: We assess mathematical reasoning capability using GSM8K (Cobbe et al., 2021), which consists of 8.5K high-quality grade school math problems. Evaluations are conducted on the test set using the LM Eval Harness framework prompting in a 5-shot setting, reporting the "exact_match, flexible-extract" metric.

- **Code Generation (HumanEval)**: We adopt HumanEval (Chen et al., 2021), comprising 164 unique programming problems, to evaluate the coding capabilities of LLMs. For chat experiments, we use the vLLM framework with the open-instruct implementation and report the pass@10 performance.

- **Medical Question Answering (MedQ)**: To assess medical knowledge, we utilize three benchmarks—PubMedQA (Jin et al., 2019), MedMCQA (Pal et al., 2022), and MedQA (Jin et al., 2021). Evaluations are performed using the LM Eval Harness framework. For PubMedQA, we report the "acc, none" metric; for MedMCQA and MedQA, we report the "acc_norm, none" metric.

- **Instruction Following (IFEval)**: We evaluate the instruction-following ability of LLMs using the IFeval benchmark. Evaluations are conducted with the LM Eval Harness implementation, and we report the "inst_level_strict_acc, none" metric.

All benchmarks—including CommonSense, GSM8K, PubMedQA, MedMCQA, MedQA, and IFeval—are evaluated using the LM Eval Harness framework (Gao et al., 2023), following their default settings unless specified otherwise.

### D.3 Hyperparameter Configurations

**Instruction fine-tuning.** In our instruction fine-tuning experiments, we follow the implementation of Ivison et al. (2023). For instruction fine-tuning, we set the maximum sequence length to 1024, the global batch size to 128, and we train the model for 2 epochs. For the Llama-2-7B model, we use a learning rate of 2e-5, with a cosine decay learning rate scheduler. The learning rate is set to 2e-5 for fine-tuning both the Llama-2-7B-Chat model on the MetaMathQA dataset and the Gemma-2B-IT model, while a learning rate of 1e-5 is used for fine-tuning the Llama-2-7B-Chat model on the PMC-LLaMA-Instruct dataset; all these settings employ a warm-up ratio of 0.03 and a cosine decay learning rate scheduler. For LoRA, we set the learning rate as 1e-4. The other hyperparameters in the experiments are as follows.

**Fine-tuning Llama-2-7B on MetaMathQA.**

- Learning rate: 2e-5.
- Update fraction of MoFO: $\alpha\% = 15\%$.
- LoRA: $r = 4, 16, 64, 256$. We report the best-performing hyperparameter configuration for the fine-tuning task in Table 1, which, in this case, is $r = 256$.

**Fine-tuning Llama-2-7B-Chat on PMC-LLaMA-Instruct.**

- Learning rate: 1e-5.
- Update fraction of MoFO: $\alpha\% = 10\%$.
- LoRA: $r = 16, 256$. We report the best-performing hyperparameter configuration for the fine-tuning task in Table 5, which, in this case, is $r = 256$.

**Fine-tuning Llama-2-7B-Chat on MetaMathQA.**

- Learning rate: 2e-5.
- Update fraction of MoFO: $\alpha\% = 15\%$.
- LoRA: $r = 16, 256$. We report the best-performing hyperparameter configuration for the fine-tuning task in Table 7, which, in this case, is $r = 256$.

**Fine-tuning Gemma-2B-IT on MetaMathQA.**

- Learning rate: 2e-5.
- Update fraction of MoFO: $\alpha\% = 5\%$.
- LoRA: $r = 16, 256, 512$. We report the best-performing hyperparameter configuration for the fine-tuning task in Table 6, which, in this case, is $r = 512$.

**Hyperparameters in the Pareto comparison.** To provide a comprehensive comparison, we explore various hyperparameter settings for $\lambda_1$, $\lambda_2$, LoRA's rank, and the update fraction $\alpha\%$ in MoFO in Figure 4. Specifically, we set $\lambda_1$ as 1e-4, 1e-5, 1e-6, 1e-7, while $\lambda_2$ is set as 1e-2, 5e-3, 1e-3, 5e-4, and 1e-4. The update fraction $\alpha\%$ in MoFO is set as $5\%$, $10\%$, $15\%$, $20\%$, $40\%$, $80\%$. The rank of LoRA is set as 4, 16, 64, 256.

**Continual fine-tuning.** In our continual fine-tuning experiments, we follow the default settings of the TRACE benchmark. We sequentially train TinyLlama-1.1B on the TRACE benchmark datasets: C-STANCE, FOMC, MeetingBank, Py150, ScienceQA, NumGLUE-cm, NumGLUE-ds, and 20Minuten for 5, 3, 7, 5, 3, 5, 5, and 7 epochs, respectively. We use a learning rate of 1e-5 with a cosine decay schedule and a batch size of 64. The parameter update fraction for MoFO is set to $5\%$.

All experiments are conducted on four A800 (80GB) GPUs.

### D.4 More Explanation on the partitioning and Calculation of distance

**Partitioning.** We use the default partitioning scheme in PyTorch's Transformer implementation. Different types of parameters within the Transformer, such as query (Q), key (K), value (V) weights

for attention heads, and feed-forward network (FFN) weights, are divided into separate partitions. Notably, in the default PyTorch implementation, within a layer, the query (Q) weights of all attention heads are grouped into a single partition. The same applies to the key (K) and value (V) weights. Our momentum-based filtering mechanism is applied to each partition individually.

**Calculation of distance.** Following the notation in Section , we suppose that the parameter parameters are partitioned into

$$\theta = (\theta^{(1)}, \theta^{(2)}, \ldots, \theta^{(B)}).$$

Denote the pre-trained model by $\theta_0$ and the fine-tuned model by $\theta$.

First, we calculate the relative change of parameters $\frac{\|\theta^{(k)} - \theta_0^{(k)}\|}{\|\theta_0^{(k)}\|}$ in each partition $k \in \{1, 2, \ldots, B\}$.

Second, we compute the distance from the pre-trained model $\theta_0$ to the fine-tuned model $\theta$ by averaging the relative changes across all partitions, defined as:

$$D(\theta, \theta_0) = \frac{1}{B} \sum_{k=1}^{B} \frac{\|\theta^{(k)} - \theta_0^{(k)}\|}{\|\theta_0^{(k)}\|}.$$

# E ADDITIONAL EXPERIMENTS

## E.1 IMPACT OF THE UPDATE FRACTION

In this section, we first investigate the impact of the update fraction of parameters in the MoFO algorithm at each iteration, and then explore the effects of different update strategies within MoFO.

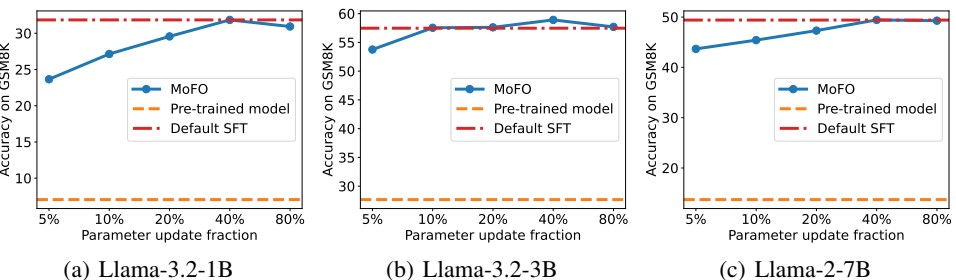

| (a) Llama-3.2-1B | (b) Llama-3.2-3B | (c) Llama-2-7B |

Figure 6: The performance of LLMs with different sizes on the math reasoning task (GSM8K) after fine-tuning on MetaMathQA using MoFO with different update fractions ($\alpha\%$) of parameters. Results show that across models of different sizes, setting the fraction $\alpha\%$ to approximately 20% allows MoFO to reach fine-tuning performance similar to the default FT (with up to 3% performance drop).

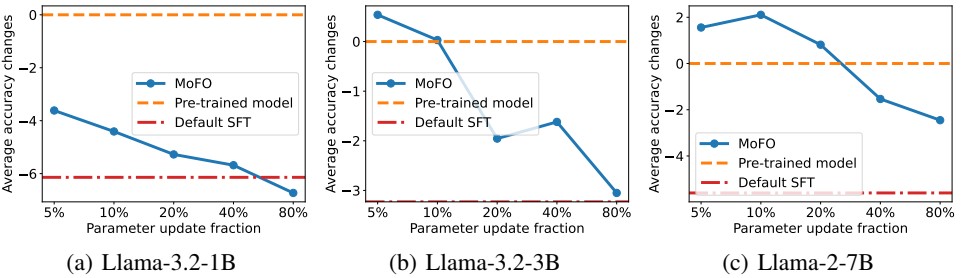

| (a) Llama-3.2-1B | (b) Llama-3.2-3B | (c) Llama-2-7B |

Figure 7: Average accuracy changes on MMLU, HumanEval, Commonsense Reasoning benchmarks compared to the pre-trained LLMs of different sizes after fine-tuning on MetaMathQA using MoFO with different update fractions ($\alpha\%$) of parameters. Larger LLMs tend to retain their pre-training knowledge more effectively when fine-tuned with MoFO, even when using smaller fractions of parameter updates.

**Impact of update fraction of parameters in MoFO.** Following the setting in Section 4.2, we fine-tune Llama-3.2-1B, Llama-3.2-3B, and Llama-2-7B on the MetaMathQA dataset using MoFO with varying update fractions of parameters at each iteration for 2 epochs. The experimental results of math reasoning (GSM8K) and average general capability performance changes are presented in Figure 6 and Figure 7.

The parameter update fraction affects the fine-tuning performance. Figure 6 shows that larger update fractions can improve MoFO's optimization effectiveness. Furthermore, in Llama-2-7B and Llama-3.2-3B, MoFO with a 5% parameter update fraction is sufficient to achieve nearly 90% of the performance of Default FT. Besides, experimental results show that setting the update fraction as $\alpha$ to approximately 20% enables MoFO to attain fine-tuning performance comparable to the default FT across various model sizes.

The parameter update fraction also affects the preservation of general capabilities. Figure 7 indicates that larger LLMs effectively maintain their pre-training knowledge when fine-tuned with MoFO, especially when using update fraction $\alpha$ less than 10%. Beyond the threshold of 20%, further increases in the parameter update fraction lead to a decline in general capabilities. Despite this, MoFO still forgets significantly less than Default FT in larger LLMs.

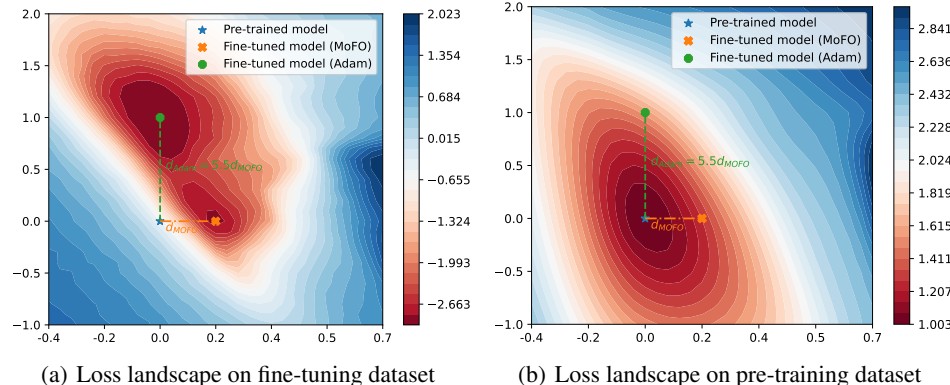

(a) Loss landscape on fine-tuning dataset          (b) Loss landscape on pre-training dataset

Figure 8: The loss landscapes of Pythia-160m after fine-tuning on a subset of the FLAN dataset using Adam optimizer and MoFO. We plot the loss landscapes on (a) the fine-tuning dataset and (b) the pre-training dataset (Pile). A logarithmic scale is applied to the loss values for better visualization. We find that MoFO, reaching a closer point to the pre-trained model, has minimal fine-tuning loss and lower pre-training loss, compared to Adam.

Table 4: Pythia-160m's performance on common sense tasks, after being fine-tuned with the Adam optimizer and MoFO. The results indicate that MoFO significantly mitigates catastrophic forgetting. Bold values denote the best results among these optimizers.

|  | HellaSwag | ARC-easy | ARC-challenge | Average |
|---|---|---|---|---|
| Pythia-160m | 30.1 | 39.6 | 23.8 | 31.2 |
| Adam | 28.3 | 37.4 | 22.1 | 29.3 |
| MoFO | **29.9** | **42.0** | **22.9** | **31.6** |

In summary, MoFO can preserve pre-training knowledge and significantly enhance fine-tuning performance by choosing a moderate update fraction, avoiding the extremes of too small or too large fractions.

### E.2 VALIDATING MoFO'S IMPACT ON PRESERVING PRE-TRAINING KNOWLEDGE THROUGH PROXIMITY

In this section, we empirically examine whether MoFO achieves its intended goal of converging to a minimum closer to the pre-trained model and mitigating forgetting mentioned in Section 3.

Our exploratory experiment shows that MoFO indeed converges to a minimum closer to the pre-training model. As shown in Figure 8(a), both MoFO and the Adam optimizer achieve minimal fine-tuning loss, indicating that switching from Adam to MoFO does not lead to performance degradation. Moreover, the distance from the pre-trained model to the minimum reached by MoFO is approximately 20% of that reached by the default Adam optimizer.

Our experiment demonstrates that the reduced parameter movement achieved by MoFO effectively mitigates the forgetting of pre-training knowledge. As shown in Figure 8(b), the fine-tuned model using MoFO experiences a smaller increase in pre-training loss. Additionally, Table 4 shows that MoFO achieves higher accuracy on commonsense reasoning tasks, indicating less forgetting.

### E.3 MORE EXPERIMENTAL RESULTS IN INSTRUCTION FINE-TUNING

**Results of fine-tuning on PMC-LLaMA-Instruct.** We fine-tune Llama-2-7B-Chat on the PMC-LLaMA-Instructions dataset using various baseline methods and present the experimental results on medical question answering (MedQ) and general capabilities in Table 5. Since the MMLU benchmark

Table 5: The performance on the fine-tuning task (medical QA task), measured by MedQ, and general capability scores of Llama-2-7B-Chat after fine-tuning on the PMC-LLaMA-Instruct dataset. The figure on the right visualizes both MedQ accuracy and general capability scores. The results show that MoFO achieves comparable performance in the MedQ while significantly mitigating forgetting of general capabilities. Bold values denote the best results among these methods.

| Method | MedQ | General Capability | | | |
|---|---|---|---|---|---|
| | | CR | IFEval | HumanEval | Avg. |
| Llama-2-7B-Chat | 49.8 | 65.6 | 41.4 | 24.3 | 43.8 |
| Default FT | 54.3 | 64.6 | 32.1 | 20.6 | 39.1 |
| HFT | **54.4** | 65.2 | 33.5 | 23.1 | 40.6 |
| LoRA | 54.2 | 64.4 | 33.9 | 23.5 | 40.6 |
| MoFO | 54.3 | **65.5** | **41.1** | **24.1** | **43.6** |

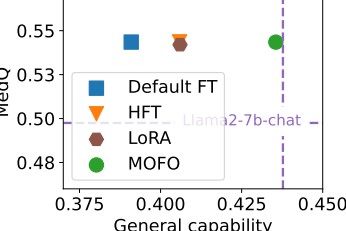

already contains medical-related instances (Hendrycks et al., 2021), which may lead to improved performance after fine-tuning, we instead use IFEval to assess general capabilities.

MoFO performs well on the fine-tuning task of medical QA. It achieves compatible performance compared to Default FT and HFT. In terms of general capabilities, MoFO demonstrates the least degradation compared to other baselines, with an average accuracy reduction of only 0.2%. Specifically, on the IFEval benchmark, our method only exhibits a minor reduction of 0.3%, while Default FT, HFT, and LoRA experience significant degradations ranging from 7.5% to 9.3%. On code generation (HumanEval) tasks and commonsense reasoning (CR) benchmarks, our method also only exhibits a minor reduction less than 0.2%.

Table 6: The performance of the fine-tuning task (math), measured by GSM8K, and the general capability scores of Gemma-2B-IT after fine-tuning on the MetaMathQA dataset. The figure on the right visualizes both GSM8K accuracy and general capability scores. The results show that MoFO achieves comparable performance in the fine-tuning task, while significantly mitigating forgetting of general capabilities. Bold values denote the best results among these methods.

| Method | GSM8K | General Capability | | | |
|---|---|---|---|---|---|
| | | CR | IFeval | HumanEval | Avg. |
| Gemma-2B-IT | 11.4 | 57.6 | 33.6 | 31.5 | 40.9 |
| Default FT | 42.0 | 52.1 | 24.3 | 20.6 | 32.3 |
| HFT | 41.5 | 53.9 | 24.1 | 21.2 | 33.1 |
| LoRA | 40.6 | 54.4 | 26.1 | **29.8** | 36.8 |
| MoFO | **42.1** | **55.0** | **28.7** | 29.1 | **37.6** |

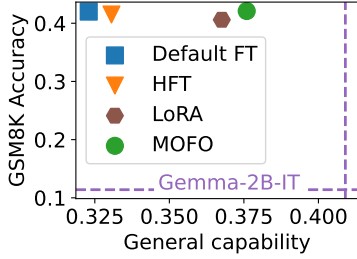

**Results of Gemma-2B-IT fine-tuning on MetaMathQA.** We also explore how MoFO performs in other LLMs. Specifically, we fine-tune Gemma-2B-IT on MetaMathQA using various baseline methods and present the experimental results on mathematical reasoning (GSM8K) and general capabilities in Table 6. The experimental results demonstrate that MoFO achieves comparable performance of the fine-tuning task to Default FT and HFT across different models. In terms of general capabilities, MoFO exhibits significantly less forgetting compared to other baselines. This result demonstrates the versatility of the MoFO algorithm.

We also fine-tune the Llama-2-7B-Chat on the MetaMathQA dataset. The results are presented in Table 7. The results demonstrate that our approach achieves performance comparable to Default FT and HFT while exhibiting less forgetting compared to baseline methods.

In summary, our MoFO algorithm shows competitive performance in instruction fine-tuning while preserving the general capabilities, effectively alleviating forgetting.

Table 7: The performance of the fine-tuning task (math), measured by GSM8K, and the general capability scores of Llama-2-7B-chat after fine-tuning on the MetaMathQA dataset. The figure on the right visualizes both GSM8K accuracy and general capability scores. The results show that MoFO achieves comparable performance in the fine-tuning task, while significantly mitigating forgetting of general capabilities. Bold values denote the best results among these methods.

| Method | GSM8K | General Capability | | | |
| --- | --- | --- | --- | --- | --- |
| | | CR | IFeval | HumanEval | Avg. |
| Llama-2-7B-Chat | 13.7 | 65.6 | 41.4 | 24.3 | 43.8 |
| Default FT | **48.4** | 62.8 | 30.7 | 15.6 | 36.4 |
| HFT | 46.9 | 63.4 | 31.8 | 20.0 | 38.4 |
| LoRA | 45.3 | 63.9 | 35.6 | 21.0 | 40.2 |
| MoFO | 47.1 | **64.0** | **37.1** | **21.7** | **40.9** |

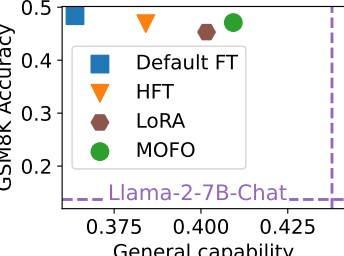

## E.4 TRANING PROCESS OF MoFO

In this subsection, we analyze the differences between the training processes of MoFO and the default SFT.

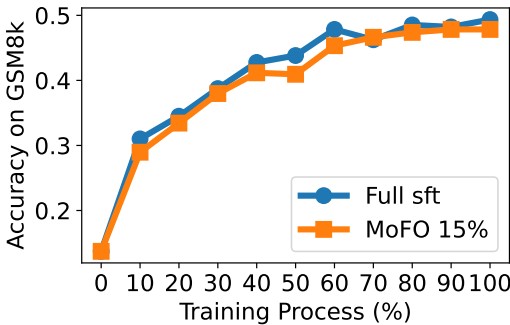

Figure 9: The GSM8K accuracy achieved during the fine-tuning of Llama-2-7B on the MetaMathQA dataset. The update fraction of MoFO is $\alpha\% = 15\%$.

Following the setting in Section 4.2, we present the GSM8K accuracy achieved during the fine-tuning of Llama-2-7B on the MetaMathQA dataset with different methods in Figure 9. The results demonstrate that the MoFO method can achieve training effectiveness comparable to the default fine-tuning approach.

## E.5 COMPARISON WITH MORE FINE-TUNING METHODS

In this subsection, we compare our proposed method with the Heterogeneous Model Averaging (HMA) (Lin et al., 2024). HMA approach evenly divides the LLM into three parts—the input part, the middle part, and the output part—and averages these parts with different ratios. To facilitate a comprehensive comparison, following the setting in Section 4.2, we evaluate the fine-tuning and forgetting mitigation performance for different HMA strategies. We select 15 different combinations of averaging ratios for different parts as follows: {(0.05, 0.2, 0.35), (0.1, 0.2, 0.3), (0.2, 0.2, 0.2), (0.3, 0.2, 0.1), (0.35, 0.2, 0.05), (0.3, 0.5, 0.7), (0.4, 0.5, 0.6), (0.5, 0.5, 0.5), (0.6, 0.5, 0.4), (0.7, 0.5, 0.3), (0.65, 0.8, 0.95), (0.7, 0.8, 0.9), (0.8, 0.8, 0.8), (0.9, 0.8, 0.7), (0.95, 0.8, 0.65)}. We plot the results to construct a Pareto front in Figure 10.

Results show that our proposed method, MoFO achieves a more effective Pareto front compared to the baselines.

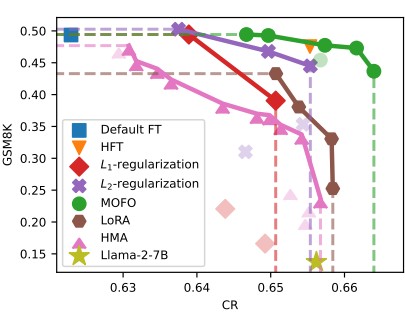

Figure 10: The performance on the math task (GSM8K) and the scores in Commonsense Reasoning of Llama-2-7B after fine-tuning on the MetaMathQA dataset. The results show that the MoFO algorithm achieves a better Pareto front. The pink triangle represents the model obtained through HMA.

