# OpenReview forum: "MoFO: Momentum-Filtered Optimizer for Mitigating Forgetting in LLM Fine-Tuning"
_ICLR.cc/2025/Conference — Submitted to ICLR 2025_

### Official Review · Reviewer_5vpM · 2024-11-01

**Soundness:** 2
**Presentation:** 3
**Contribution:** 2
**Rating:** 5
**Confidence:** 3

**Summary:**

The motivation of this paper is to address the challenge of forgetting in large language models (LLMs) during fine-tuning. The main contribution is the proposed Momentum-Filtered Optimizer (MoFO), a fine-tuning algorithm based on block coordinate descent that selectively updates high-momentum parameters to prevent forgetting.

**Strengths:**

(a) The paper presents a convergence analysis of the proposed optimizer.

(b) It includes experiments on a diverse set of LLM models and a range of advanced NLP fine-tuning tasks.

**Weaknesses:**

(a) Unclear Motivation: In the Introduction (Lines 54-55), the statement “However, regularization-based methods alter the original (fine-tuning) loss function, which can potentially affect the model’s performance on fine-tuning tasks” lacks citations or experimental support, weakening the rationale for excluding regularization-based, replay-free fine-tuning methods as baselines. Providing examples or references of such methods that negatively impact fine-tuning performance would help clarify this claim. Additionally, explaining how modifications to the “original” loss function might specifically affect performance would strengthen the motivation for a regularization-free approach. The approach of keeping fine-tuned model parameters close to pre-trained models to prevent forgetting is already well-established in continual learning [1], limiting the novelty of this finding.

(b) Insufficient Baselines: The exclusion of regularization-based fine-tuning methods as baselines lacks sufficient justification. Adding comparisons with well-known regularization-based approaches, such as proximal gradient and EWC-based methods [1], would provide a more thorough evaluation. Furthermore, the current baselines mainly focus on parameter-efficient fine-tuning techniques, omitting full fine-tuning methods relevant for a comprehensive comparison in the LLM context. Including full fine-tuning baselines, as shown in recent work on LLMs [2], would enhance the fairness and depth of the comparative analysis.

(c) Lack of Clear Novelty and Contribution: The specific contributions and novelty of the proposed method in addressing forgetting in fine-tuning LLMs are difficult to discern, as the forgetting issue is not unique to LLMs and has been explored in other settings. Clarifying how this approach differs from or builds upon existing forgetting mitigation techniques, especially in the LLM context, would make its contributions more apparent.

(d) Minor Issues: The use of non-integer epochs is unclear (Line 103); specifying iterations would be more appropriate. Additionally, please clarify the batch size and learning rate used in each experiment, as these details are essential for reproducibility.

[1] Kirkpatrick, J., et al. “Overcoming catastrophic forgetting in neural networks.” Proceedings of the National Academy of Sciences 114.13 (2017): 3521-3526.

[2] Lv, K., et al. “Full parameter fine-tuning for large language models with limited resources.” arXiv preprint arXiv:2306.09782 (2023).

**Questions:**

See weakness

---

> ### Author Response · Authors · 2024-11-25
> **Response (Part I)**
>
> ## **Weakness (a)**
> *Unclear Motivation: In the Introduction (Lines 54-55), the statement “However, regularization-based methods alter the original (fine-tuning) loss function, which can potentially affect the model’s performance on fine-tuning tasks” lacks citations or experimental support, weakening the rationale for excluding regularization-based, replay-free fine-tuning methods as baselines. Providing examples or references of such methods that negatively impact fine-tuning performance would help clarify this claim. Additionally, explaining how modifications to the “original” loss function might specifically affect performance would strengthen the motivation for a regularization-free approach. The approach of keeping fine-tuned model parameters close to pre-trained models to prevent forgetting is already well-established in continual learning [1], limiting the novelty of this finding.*
>
> ## **Response to Weakness (a)**
>
> Thanks for your feedback. We have added some references to explain that regularization-based methods may negatively impact fine-tuning performance. Moreover, we conduct a preliminary theoretical analysis of the potential adverse effects of altering the original loss function. We believe these can serve as motivation for proposing a regularization-free method.
>
> ### **References on the negative impact of regularization-based methods:**
>
> Thanks for your comments. We have added the following reference into our literature review section.
>
> [1][2][3][4] have pointed out some limitations of regularization-based approaches. They may exhibit poor adaptability to new tasks in long sequential learning.
>
> ### **Novelty of the Motivation of our methods:**
> Thanks for your comments.
>
> Although regularization methods and our MoFO share the high-level motivation of "keeping fine-tuned model parameters close to pre-trained models to prevent forgetting", they differ in the specific motivations on their proposed methods. While regularization-based methods make changes to the loss function **formulation**, our method introduces changes at the level of the **optimization algorithm** of the loss function.
>
> We can analogize the above discussion to previous work on sparse optimization [5][6]. Consider an optimization problem
> $$
> \min_{x \in \mathbb{R}^n} f(x).
> $$
> We aim to find a solution with few nonzero components. Two basic methods can be implemented to solve this:
> - The $\ell_1$-regularization method promotes sparsity in the solution by **modifying the problem formulation**. Specifically, it adds an $\ell_1$-norm penalty to the objective function, i.e.,
> $$
> \min_{x \in \mathbb{R}^n} f(x) + \lambda \Vert x \Vert_1.
> $$
> - The active set method works on the **optimization algorithm**. It iteratively identifies and optimizes over a subset of variables expected to be non-zero, maintaining sparsity by only updating solutions within this active set at each step.
>
> Both methods share the primary motivation of finding sparse optimal solutions, but they differ in the motivations behind their design.
>
> **References:**
>
> [1] Aljundi, Rahaf, et al. "Memory aware synapses: Learning what (not) to forget." Proceedings of the European conference on computer vision (ECCV). 2018.
>
> [2] Panda, Ashwinee, et al. "Lottery ticket adaptation: Mitigating destructive interference in llms." arXiv preprint arXiv:2406.16797 (2024).
>
> [3] Lesort, Timothée, Andrei Stoian, and David Filliat. "Regularization shortcomings for continual learning." arXiv preprint arXiv:1912.03049 (2019).
>
> [4] Wu, Tongtong, et al. "Pretrained language model in continual learning: A comparative study." International Conference on Learning Representations 2022. OpenReview, 2022.
>
> [5] Nocedal, Jorge, and Stephen J. Wright, eds. Numerical optimization. New York, NY: Springer New York, 1999.
>
> [6] Boyd, Stephen, and Lieven Vandenberghe. Convex optimization. Cambridge university press, 2004.

---

> ### Author Response · Authors · 2024-11-25
> **Response (Part II) (Response to Weakness (a), Cont'd)**
>
> ### **Negative Impact on Modifying the Original Loss Function:**
>
> Although both MoFO and regularization methods aim to prevent forgetting by keeping the fine-tuned model parameters close to the pre-trained model, MoFO also considers the optimization perspective. We provide a theoretical analysis demonstrating how modifying the original loss function can impact convergence to minima.
>
> **Theoretical Analysis**
>
> Suppose that the pre-trained model is $\theta_0$. Denote the pre-training loss function by $\mathcal{L}_{\rm pretrain}(\cdot)$ and the fine-tuning loss function by $\mathcal{L}(\cdot)$.
>
> Suppose that we add a differentiable regularizer $\mathcal{R}(\theta)$ to the fine-tuning loss, i.e.,
> $$
> \mathcal{L}_{\rm reg}(\theta) := \mathcal{L}(\theta) + \lambda \mathcal{R}(\theta).
> $$
> Here, $\lambda > 0$ is the regularization hyper-parameter.
>
> **Assumption 1**:
> - $\mathcal{R}(\theta) \geq 0$.
> - $\mathcal{R}(\theta_0) = 0$.
> - If $\theta \neq \theta_0$, then $\nabla \mathcal{R}(\theta) \neq 0$.
>
> Assumption 1 is a very general and natural assumption. Assumption 1 holds for L2 regularizer $\mathcal{R}(\theta) = \frac{1}{2} \Vert \theta - \theta_0 \Vert_2^2$, L1 regularizer $\mathcal{R}(\theta) = \Vert \theta - \theta_0 \Vert_1$, and the quadratic form regularizer $\mathcal{R}(\theta) = \frac{1}{2} (\theta - \theta_0)^\top P (\theta - \theta_0)$, where $P$ is positive definite.
>
> **Assumption 2**: The fine-tuning loss $\mathcal{L}(\cdot)$ is smooth. The pre-trained model is not a minimum if the fine-tuning loss $\mathcal{L}(\cdot)$, i.e.,
> $$
> \theta_0 \notin \arg\min_{\theta \in \mathbb{R}^d} \mathcal{L}(\theta).
> $$
>
> Assumption 2 is natural, because if the pre-training state has already been the minimum of fine-tuning loss, then there is no need to do the fine-tuning.
>
> **Theorem 1**: Under Assumption 1 and 2, for any minimum $\theta_{\rm reg}$ of the regularization loss $\mathcal{L}_ {\rm reg}(\cdot)$, $\theta_{\rm reg}$ cannot be a minimum of the fine-tuning loss $\mathcal{L}(\cdot)$.
>
> **Proof**:
>
> **Case 1**: $\theta_{\rm reg} = \theta_0$.
> By Assumption 2, we have
> $$
> \theta_{\rm reg} = \theta_0 \notin \arg\min_{\theta \in \mathbb{R}^d} \mathcal{L} (\theta). \quad (1)
> $$
>
> Theorem 1 holds for Case 1.
>
> **Case 2**: $\theta_{\rm reg} \neq \theta_0$.
> $\theta_{\rm reg}$ is a minimum of $\mathcal{L}_ {\rm reg}(\cdot)$, so it is a stationary point of $\mathcal{L}_ {\rm reg}(\cdot)$, i.e.,
> $$
> \nabla \mathcal{L}_ {\rm reg}(\theta_ {\rm reg}) = \nabla \mathcal{L}(\theta_ {\rm reg}) + \lambda \nabla \mathcal{R}(\theta_ {\rm reg}) = 0.
> $$
> Since $\theta_{\rm reg} \neq \theta_0$, by Assumption 2 and (1),  $\nabla \mathcal{R}(\theta_{\rm reg}) \neq 0$. So, $\nabla \mathcal{L}(\theta_{\rm reg}) \neq 0$. Thus, $\theta_{\rm reg}$ is not a stationary point and is not a minima of the fine-tuning loss $\mathcal{L}$.

---

> ### Author Response · Authors · 2024-11-25
> **Response (Part III)**
>
> ## **Weakness (b)**
> *Insufficient Baselines: The exclusion of regularization-based fine-tuning methods as baselines lacks sufficient justification. Adding comparisons with well-known regularization-based approaches, such as proximal gradient and EWC-based methods [7], would provide a more thorough evaluation. Furthermore, the current baselines mainly focus on parameter-efficient fine-tuning techniques, omitting full fine-tuning methods relevant for a comprehensive comparison in the LLM context. Including full fine-tuning baselines, as shown in recent work on LLMs [8], would enhance the fairness and depth of the comparative analysis.*
>
> ## **Response to Weakness (b)**
>
> Thanks for your advice. We have added some baselines for both continual learning and LLM fine-tuning.
>
> ### **Regularization Baselines in Continual Learning**
>
> Following your suggestion, in addition to the original baselines, we have included two regularization-based methods, EWC [7] and proximal gradient descent (proximal GD), as baselines, in continual learning experiments of Section 4.3. To present our results more clearly, we have reorganized our findings into a comparison table and two orthogonal combination tables. (In these tables, OP and BWT are two performance metrics of continual learning, standing for overall performance and BackWard Transfer, respectively. Higher OP and BWT values reflect better mitigation of forgetting.)
>
> **Table 1: Performance Comparison of MoFO with regularization-based methods**
>
> | Method       | OP   | BWT   |
> |--------------|------|-------|
> | Default FT   | 38.4 | -10.3 |
> | MoFO ($\alpha$%=5%) | **41.3** | **-5.4**  |
> | EWC          | 41.1 | -8.3  |
> | Proximal GD      | 38.2 | -11.2 |
>
>
> **Table 2: Orthogonal Combination**
> | Method            | OP   | BWT   |
> |-------------------|------|-------|
> | EWC               | 41.1 | -8.3  |
> | MoFO ($\alpha$%=5%)  | 41.3 | -5.4  |
> | EWC + MoFO        | **43.2** | **-4.4**  |
>
> From these results, we observe that MoFO not only outperforms other methods when used alone but also provides additional benefits when combined orthogonally. We believe these additions strengthen the validity of our experimental results.
>
> ### **Clarification on Full Fine-Tuning Baselines**
>
> We clarify that we are NOT comparing to PEFT only method, LoRA. We have also included HFT, L1 and L2 regularizations as our baselines. In the "Comparison from a Pareto Perspective" part in Section 4.2 of our manuscript, we incorporate L1 and L2 regularizations, which serve as full-parameter fine-tuning baselines.
>
> Potential confusion: Although MoFO and HFT update only a subset of parameters at each iteration, they **do not** restrict the parameter space throughout the entire fine-tuning process. Therefore, they are generally NOT classified into parameter-efficient fine-tuning (PEFT) techniques.
>
> Following your recommendation, we conduct experimental comparisons with the full fine-tuning baselines you have mentioned in [8] (LOMO), following the setting in Table 1 of Section 4. As LOMO is the low-memory implementation of Adam/SGD without compromising performance, we compare MoFO with the two baselines in [8], SGD and Adam.
>
> Note that Adam is exactly the Default FT method in our experiment. Below we provide additional comparison between MoFO and SGD.
>
> | Method                | GSM8K (FT) |       | Avg. of General Capacities |       | CR (General)  | MMLU (General) | HumanEval (General) |
> |-----------------------|------------|-------|----------------------------|-------|---------------|----------------|---------------------|
> | Default FT (Adam)     | **49.4**   |       | 38.3                       |       | 62.3          | 36.6           | 16.1                |
> | MoFO ($\alpha$%=15%)| 47.7       |       | **44.3**                   |       | 65.7      | 42.7       | 24.6            |
> | SGD                   | 25.8       |       | 39.9                       |       | 64.3          | 31.0           | 24.4                |
>
>
> The experimental results demonstrate that MoFO significantly outperforms SGD in both forgetting mitigation and fine-tuning performance.
>
>
> **References:**
>
> [7] Kirkpatrick, J., et al. “Overcoming catastrophic forgetting in neural networks.” Proceedings of the National Academy of Sciences 114.13 (2017): 3521-3526.
>
> [8] Lv, K., et al. “Full parameter fine-tuning for large language models with limited resources.” arXiv preprint arXiv:2306.09782 (2023).

---

> ### Author Response · Authors · 2024-11-25
> **Response (Part IV)**
>
> ## **Weakness \(c)**
> *Lack of Clear Novelty and Contribution: The specific contributions and novelty of the proposed method in addressing forgetting in fine-tuning LLMs are difficult to discern, as the forgetting issue is not unique to LLMs and has been explored in other settings. Clarifying how this approach differs from or builds upon existing forgetting mitigation techniques, especially in the LLM context, would make its contributions more apparent.*
>
> ## **Response to Weakness \(c)**
> Thanks a lot for your suggestion. We agree that "forgetting issue is not unique to LLMs and has been explored in other settings". To clarify our contribution and novelty, we'd like to highlight two special aspects of LLM: (i) lack of pretraining data for replay; (ii) dominance of Adam algorithm.
>
> More specifically, our contributions and novelty can be understood from three perspectives.
>
> **Setting:** Our work differs from earlier continual learning studies by focusing on the no-replay-data scenario in LLMs, which has been less explored in existing research.
>
> **Methodology:** We adopt the principle that "less distance may imply less forgetting" from continual learning. However, we diverge from traditional regularization-based approaches by employing an algorithm-modification strategy. Specifically, we leverage the implicit bias of Block Coordinate Descent (BCD) to achieve smaller parameter distances, thereby reducing forgetting without relying on regularization techniques.
>
> **Algorithm Design:** Given that Adam is the default optimizer for LLMs, we have designed a novel algorithm based on Adam and rigorously proved its convergence. This tailored algorithm ensures compatibility with existing LLM training processes.
>
> We'll explain in detail below, and we will add the explanations to the modified version.
>
> ---
> ### **(A). No-Replay-Data Setting**
>
> **(A.1). New challenge: "rare replay pre-training data" in LLM fine-tuning**:
>
> The first notable aspect of LLM development is the scarcity of "replay data", which is the pretraining data used for training the LLM. For instance, the pretraining dataset for open-source LLMs like LLama and Mistral and closed-source LLMs such as GPT-4 and Claude are NOT available. This setting is rarely explored within the realm of classical continual learning. To be fair, there exist LLMs whose accompanying pretraining-data is released, but this situation is rare, and for most (if not all) popular LLMs the replay data is not available. Thus, "no replay data" is a new setting in LLM era, posing new challenge for continual learning.
>
> **(A.2). Our Contriubtion: Algorithm Design for the No-Replay-Data Setting in LLM fine-tuning**:
>
>    In the era of traditional continual learning, replay-based methods and regularization-based methods have been prevalent. Not only the class of replay-based algorithms use replay data, even some regularization-based methods also use information from the pretraining data (e.g. EWC uses Hessian computed based on pretraining data). Thus, in the no-replay-data setting, many of these classical methods are no longer applicable. This raises a question: what else can we explore for the no-replay setting?
>
> **One of our contributions is to design forgetting mitigation algorithm for no-replay-data setting.** The designed algorithm surely can be used for all continual learning setting, but especailly useful for LLMs with no replay data.
>
> ---
> ### **(B) Keeping-Formulation-Modifying-Algorithm Approach: Using BCD to Achieve Less Distance**
> We borrow the idea of "less distance may lead to less forgetting" in classical continual learning, which led to regularization-based methods in classical continual learning literature. But we notice that regularization-based methods is just a penalty-function-approach from optimization perspetive.
>
> **Our contribution**: We divert from this classical idea; instead, we notice that modifying algorithm can also achieve less distance. In particular, we realize that the classical algorithm Block Coordinate Descent (BCD) enjoys the hidden benefit of moving less distance. Using the modern language of machine learning, BCD has an implicit bias towards the less-distance solution.
>
> **Implication**: Our paper promotes a research path for the no-replay-data setting: modifying the optimization algorithm itself. This direction is less significant when replay is allowed, but its importance increases substantially with the absence of replay data. It is a promising path in mitigating LLM forgetting. Our paper makes contribution in this direction, thus is different from many classical continual learning works.

---

> ### Author Response · Authors · 2024-11-25
> **Response (Part V) (Response to Weakness (c), Cont'd)**
>
> ### **\(C) Algorithm-Design: Handling Adam and BCD**
> The second notable aspect of LLMs is the prevalence of Adam, making the design of new optimization algorithms more challenging.
>
> **(C.1). Background Knowledge: Predominance of Adam for LLM:**
>
> Earlier continual learning research [9][10][11] often focused on tasks like computer vision. In these tasks, stochastic gradient descent (SGD) is often the default optimizer. **In contrast**, for LLMs people often use the Adam optimizer. Recent studies provided some theoretical justifications why Adam outperforms SGD for Transformers [12].
>
>
> **(C.2). Adapting Optimization Algorithms for LLMs Using Adam**:
>
> Since Adam is the default choice for LLMs, new optimizer design for LLMs would need to be based on Adam instead of SGD. The complexity of Adam introduces challenges in the optimizer design:
>
> a) **Design**: Due to Adam's intricate update rules, it was not clear a priori whether modifying some parts of Adam will improve the performance. This requires some intuition of Adam's working mechanism and some experimental exploration.
>
> b) **Convergence Analysis**: Modifying algorithms within the Adam framework complicates the analysis of their convergence properties.
>
> Therefore, even though we had the idea of incorporating Block Coordinate Descent (BCD) into Adam, combining BCD with Adam is not straightforward. To design a good algorithm, we make the following efforts:
>
> a) **Identifying a Proper Combination Method**: We notice that there are multiple possible ways to combine BCD with Adam, including momentum-filtered, gradient-filtered, and MV-filtered approaches. In Section 4.4, we perform ablation studies to evaluate various selection rules, and identify momentum-filtered rule, referred to as MoFO, as the best-performing option.
>
> b) **Providing Convergence Analysis**: Leveraging recent advancements in the theoretical understanding of Adam's convergence [13], we offer a comprehensive convergence analysis for our proposed algorithm. Note that the convergence analysis of Adam is only available in the past few years, thus the convergence anlaysis of Adam-variant does not exist in continual learning literature.
>
>
>
> **References:**
>
> [9] Lopez-Paz, David, and Marc'Aurelio Ranzato. "Gradient episodic memory for continual learning." Advances in neural information processing systems 30 (2017).
>
> [10] Kirkpatrick, James, et al. "Overcoming catastrophic forgetting in neural networks." Proceedings of the national academy of sciences 114.13 (2017): 3521-3526.
>
> [11] Farajtabar, Mehrdad, et al. "Orthogonal gradient descent for continual learning." International Conference on Artificial Intelligence and Statistics. PMLR, 2020.
>
> [12] Zhang, Yushun, et al. "Why Transformers Need Adam: A Hessian Perspective." *arXiv preprint arXiv:2402.16788* (2024).
>
> [13] Shi, Naichen, et al. "Rmsprop converges with proper hyperparameter." International conference on learning representation. 2021.

---

> ### Author Response · Authors · 2024-11-25
> **Response (Part VI)**
>
> ## **Weakness (d)**
> *Minor Issues: The use of non-integer epochs is unclear (Line 103); specifying iterations would be more appropriate. Additionally, please clarify the batch size and learning rate used in each experiment, as these details are essential for reproducibility.*
>
> ## **Response to Weakness (d)**
>
> Thank you for your suggestion. In Line 103 of the original manuscript (highlighted in red in Line 155 of the revised version), we have supplemented the description of the experiments corresponding to Figure 1. Each epoch consists of approximately 309 steps. For other hyperparameter details such as learning rate, batch sizes, etc., see Appendix D.3.
>
> Regarding the hyperparameter details for all experiments presented in this paper, due to space limitations, we had already included this information in Appendix D.3 of the original manuscript. We believe these details can assist readers in reproducing the experimental results.

---

> ### Comment · Reviewer_5vpM · 2024-11-27
>
> Thank you for addressing some of my concerns. However, my major concerns remain unresolved, and I detail them below:
>
> W1:
>
> W1.1
>
> I still find the following statement from the Introduction Section problematic due to lack of rigor:
> > “However, regularization-based methods alter the original (fine-tuning) loss function, which can potentially affect the model’s performance on fine-tuning tasks.”
>
> Let me break this statement into specific issues:
> 1. Is the claim true for all regularization-based methods?
>
> The statement implies that all regularization-based methods alter the original loss function, which is not universally true. While explicit methods (e.g., L2 regularization, Elastic Weight Consolidation) indeed modify the loss function, implicit methods such as dropout, batch normalization, or early stopping do not necessarily do so in the same manner. A more rigorous formulation should specify that this applies to some regularization-based methods, with representative examples provided.
>
> 2. “Can potentially affect” is vague
>
> The phrase “can potentially affect” lacks specificity. Does this imply that performance improves, degrades, or varies unpredictably? A more precise description would help clarify the impact being discussed.
>
> 3. Ambiguity of “performance”
>
> The term “performance” is not sufficiently defined. It could refer to generalization, training loss, or others. Based on the theoretical analysis provided, it seems the focus is on the training loss rather than generalization.
>
> W1.2
>
> After reviewing the response, I am still unclear on what specific limitations of regularization-based methods motivated the development of the proposed “regularization-free regularization” approach. The rebuttal does not provide a concise explanation of this aspect. Clearly identifying these limitations is critical for establishing the motivation behind the work.
>
> W1.3
>
> The rebuttal does not adequately address my concern about the novelty of the findings. For example, the following remains unclarified:
> > “The approach of keeping fine-tuned model parameters close to pre-trained models to prevent forgetting is already well-established in continual learning.”
>
> W2:
>
> I could not locate the newly added baselines (“two regularization-based methods, EWC and proximal gradient descent (proximal GD)”) in the revised Section 4.3. There appear to be no modifications reflecting these methods in the corresponding experimental results. If these baselines were indeed added, their inclusion and results should be explicitly clarified and discussed in Section 4.3.
>
> Additionally, while I appreciate the new inclusion of results for LOMO, I suggest integrating these results into the main experiments section. Doing so would clarify the connection between LOMO and the proposed method and help to contextualize the paper’s focused setting.

---

> > ### Author Response · Authors · 2024-11-30
> > **2nd Round Response**
> >
> > ## **Response to W1.1**
> > Thank you for your detailed comment. According to your suggestion, **we have removed the mentioned sentence from the introduction** and revised the paragraphs related to our motivation (see follow-up responses).
> > We agree with your comment that while the claim is true for a portion of regularization-based methods, such as EWC [1], but some CL methods  that implement implicit regularization [2] do not explicitly alter the loss function. We have removed the original sentence and modified similar vague terms in our revised introduction.
> >
> >
> > **Reference:**
> >
> > [1] Kirkpatrick, James, et al. "Overcoming catastrophic forgetting in neural networks." Proceedings of the national academy of sciences 114.13 (2017): 3521-3526.
> >
> > [2] Bhat, Prashant, et al. "IMEX-Reg: Implicit-Explicit Regularization in the Function Space for Continual Learning." arXiv preprint arXiv:2404.18161 (2024).
> >
> > ---
> > ## **Response to W1.2**
> >
> > Thanks for your comment. In the revised introduction, we no longer use "limitations of regularization-based methods" as the motivation for proposing the "regularization-free" approach. Instead, we clarify our motivation as:
> >
> > 1. In mitigating forgetting during LLM fine-tuning, the no-replay-data setting (where no pre-training data is available) is a scenario worthy of research.
> > 2. Building upon the insight--"the closer to the previous model, the less forgetting occurs", we try to develop an optimization algorithm that converges closer and does not require pretraining data. It motivates us to adopt BCD methods in the LLM fine-tuning.
> >
> > **Please see the introduction in the revised manuscript for detailed modificiations.**
> >
> > Furthermore, our method is orthogonal to some regularization method. Our experiments in Section 4.3 demonstrate that combining EWC yields improved performance.
> >
> > ---
> > ## **Response to W1.3**
> >
> > Thank you for the comment. To clarify, we do adopt the well-established idea that "closer solutons forget less". We did not intend to claim this idea to be our novelty; instead our novelty is how to implement this idea (as explained next). We did empirically validate this idea in LLM (Section 3): solutions by different fine-tuning algorithms differ significantly in distance, leading to different performance in forgetting. This validation is not surprising for continual learning researchers, but we think it is a necessary step for the development of the optmization algorithm. We include this validation for completeness of the story.
> >
> > To avoid such confusion, we have revised the corresponding part in the introduction, as quoted below:
> >
> > _"We adopt an important insight in continual learning: the closer to the previous model, the less forgetting occurs."_
> >
> > ### **We summarize our novelty as follows:**
> > - **We propose to use BCD to achieve the idea "the closer, the less forgetting"**. Specifically, we ask: what optimization methods inherently move only small distances from the initial point? Classical BCD methods emerge as strong candidates, as they update only a subset of parameters during each iteration, thereby are implicitly biased toward solutions closer to the initial model.
> > - **We successfully combine Adam with BCD while preserving both fine-tuning performance and the theoretical convergence guarantees of Adam.** Incorporating BCD into LLM fine-tuning is a challenging task. This is primarily because Adam, the predominant optimizer for LLM training, differs substantially from SGD, which has been extensively studied in earlier continual learning works. It complicates both optimizer design and convergence analysis. To resolve the challenges, MoFO selects the most effective parameters at each iteration—those with large momentum magnitudes for reducing the fine-tuning loss. MoFO only modifies the optimizer without requiring pretraining data, which helps achieve its efficiency and effectiveness during fine-tuning. Furthermore, we provide a rigorous convergence analysis of MoFO, demonstrating that its design retains the convergence properties of Adam.
> >
> > A more detailed explanation of our novelty and contributions can be found in our previous response to Weakness \(c).
> >
> > ---
> > ## **Response to W2**
> >
> > Thanks for your advice. In the newly revised manuscript, we further add the baselines of EWC/proximal GD into the Table 2 in Section 4.3, and the results for LOMO into the Table 1 in Section 4.2. We also add the discussions accordingly. These updates should now be visible in the manuscript.

---

> ### Author Response · Authors · 2024-12-03
> **Seeking for your Valuable Feedback of our 2nd Round Response**
>
> Dear Reviewer 5vpM,
>
> We wish to express our gratitude for your dedicated time and insightful comments. As the discussion phase is nearing its end, we are eagerly awaiting your valuable feedback regarding the points we addressed in the 2nd round response.  We would love to receive feedback from you. If your concern is addressed, we would be deeply grateful if you might consider providing further comments or suggestions to guide us. Your support is deeply appreciated!
>
> Sincerely,
> Authors

---

### Official Review · Reviewer_xNNU · 2024-11-03

**Soundness:** 3
**Presentation:** 2
**Contribution:** 3
**Rating:** 6
**Confidence:** 3

**Summary:**

This paper focuses on the forgetting problem of LLM fine-tuning. It proposes a new selective-updating mechanism to mitigate forgetting in the original data domain.

**Strengths:**

1. The paper works on an intriguing and important problem, which focuses on mitigating the forgetting problem in LLM fine-tuning.
2. The paper demonstrates its performance on multiple datasets and tasks.

**Weaknesses:**

1. There is no related work section in the main paper, which makes it difficult for readers to quickly follow up on the relevant topic. I suggest authors add this part in their revised version (not in Appendix).
2. The presentation of some sections is not very clear, such as how the authors calculate the distance of different model states in Section 2.2 and how they make the parameter partition in Section 2.3.
3. I am a little bit confused about some parts of the experiment setting, please refer to **Questions**.

**Questions:**

1. In Section 2.1 and Figure 2, how do you exactly obtain the distance between the original pre-trained state and fine-tuning model's state? Also, could you explain your experiment setting here, such as did you fine-tune all parameters here?
2. Based on your findings where **models that remain closer to their pre-trained state tend to preserve more pre-training knowledge**. A naive solution based on your findings is to adopt the PEFT fine-tuning methods, which only train a small part of model parameters, such as only tuning the parameters of the attention layer of an LLM. Why can we not directly use them to mitigate the forgetting problem?
3. In Section 2.2, can you detail your parameter partition algorithm, especially under the scope of LLMs?
4. In the experiment setting, is it fair to directly compare LoRA with the fully fine-tuned MoFO? It looks like a more fair comparison is to set them under the same trainable parameters set, like comparing LoRA with MoFO(PEFT version), and comparing Default FT with MoFO(FT version).

---

> ### Author Response · Authors · 2024-11-25
> **Response (Part I)**
>
> ## **Weakness 1**
> *There is no related work section in the main paper, which makes it difficult for readers to quickly follow up on the relevant topic. I suggest authors add this part in their revised version (not in Appendix).*
>
> ## **Response to Weakness 1:**
> Thank you for your valuable suggestion. We have added a related work section (Section 2) to the revised manuscript to provide a comprehensive overview of relevant studies and help readers follow up on the topic more easily.
>
> ---
> ## **Weakness 2**
> *The presentation of some sections is not very clear, such as how the authors calculate the distance of different model states in Section 2.2 and how they make the parameter partition in Section 2.3.*
>
> ## **Response to Weakness 2:**
> We would like to clarify these aspects in this response.
>
> **Parameter partitioning:** We use the default partitioning scheme in PyTorch's Transformer implementation. Different types of parameters within the Transformer, such as query (Q), key (K), value (V) weights for attention heads, and feed-forward network (FFN) weights, are divided into separate partitions. Notably, in the default PyTorch implementation, within a layer, the query (Q) weights of all attention heads are grouped into a single partition. The same applies to the key (K) and value (V) weights. Our momentum-based filtering mechanism is applied to each partition individually.
>
>
> **Calculation of distance:** Following the notation in Section 3.2 of our revised manuscript (Section 2.2 of the original manuscript), we suppose that the parameter parameters are partitioned into
> $$
> \theta = (\theta^{(1)}, \theta^{(2)}, \dots, \theta^{(B)}).
> $$
> Denote the pre-trained model by $\theta_0$ and the fine-tuned model by $\theta$.
>
> First, we calculate the relative change of parameters $\frac{\Vert \theta^{(k)} - \theta_0^{(k)} \Vert}{\Vert \theta_0^{(k)} \Vert}$ in each partition $k \in \\{1,2,\dots, B\\}$. Second, we compute the distance from the pre-trained model $\theta_0$ to the fine-tuned model $\theta$ by averaging the relative changes across all partitions, defined as:
> $$
> D(\theta, \theta_0) = \frac{1}{B} \sum_{k=1}^B \frac{\Vert \theta^{(k)} - \theta^{(k)}_0 \Vert}{\Vert \theta^{(k)}_0 \Vert}.
> $$
>
> We also add the explanations in Appendix D.4 of the revised manuscript.
>
> ---
> ## **Weakness 3**
> *I am a little bit confused about some parts of the experiment setting, please refer to Questions.*
>
> ## **Response to Weakness 3:**
>
> See our responses to your questions.
>
> ---
> ## **Question 1**
> *In Section 2.1 and Figure 2, how do you exactly obtain the distance between the original pre-trained state and fine-tuning model's state? Also, could you explain your experiment setting here, such as did you fine-tune all parameters here?*
>
> ## **Response to Question 1:**
>
> For the distance from the pre-trained model state to the fine-tuned model state, see our response to Weakness 2.
>
> For the experiments in **Section 3.1 of the revised manuscript** (Section 2.1 and Figure 2 of the original manuscript), we compare three optimization methods: Adam, Lion, and our MoFO. **We indeed fine-tune all parameters for all three methods**. Although MOFO updates only a subset of the parameters at each iteration, all the parameters are trainable during fine-tuning. In this experimental setting, we employ a batch size of 128, a learning rate of 2e-5, and a maximum sequence length of 1024. We adopt a cosine scheduler and executed approximately 309 steps per epoch. Detailed information regarding these settings is provided in Appendix D.3.

---

> ### Author Response · Authors · 2024-11-25
> **Response (Part II)**
>
> ## **Question 2**
> *Based on your findings where models that remain closer to their pre-trained state tend to preserve more pre-training knowledge. A naive solution based on your findings is to adopt the PEFT fine-tuning methods, which only train a small part of model parameters, such as only tuning the parameters of the attention layer of an LLM. Why can we not directly use them to mitigate the forgetting problem?*
>
> ## **Response to Question 2:**
>
> Although PEFT methods update only a small number of parameters to preserve more pre-training knowledge, restricting training to a subset of parameters might reduce the model's expressivity, leading to poor adaptability to fine-tuning tasks. For example, LoRA restricts the parameter change in a low-rank subspace. [1] points out that "**LoRA Learns Less and Forgets Less**".
>
> Additionally, we conduct experiments by fine-tuning only the attention layers, following the setting in Table 1 of Section 4 in the revised manuscript. While both MoFO and fine-tuning with only attention layers achieve comparable performance in alleviating forgetting, MoFO achieves better performance in finetuning task (math benchmark, GSM8K).
>
> | Method                    | GSM8K (FT) |       | Avg. of General Capacities |       | CR (General)    | MMLU (General)   | HumanEval (General) |
> |---------------------------|------------|-------|----------------------------|-------|-----------------|------------------|---------------------|
> | MoFO                      | **47.7**   |       | **44.3**                   |       | 65.7            | 42.7         | 24.6            |
> | Default FT with Attention | 44.7       |       | **44.3**                   |       | 66.5        | 42.6             | 23.7                |
>
>
> **Reference:**
>
> [1] Biderman, Dan, et al. "Lora learns less and forgets less." arXiv preprint arXiv:2405.09673 (2024).
>
> ---
> ## **Question 3**
> *In Section 2.2, can you detail your parameter partition algorithm, especially under the scope of LLMs?*
>
>
> ## **Response to Question 3:**
>
> **Parameter partitioning: (We have responded this question in Weakness 2, but we rewrite it here for your convenience of reviewing.)** We use the default partitioning scheme in PyTorch's Transformer implementation. Different types of parameters within the Transformer, such as query (Q), key (K), value (V) weights for attention heads, and feed-forward network (FFN) weights, are divided into separate partitions. Notably, in the default PyTorch implementation, within a layer, the query (Q) weights of all attention heads are grouped into a single partition. The same applies to the key (K) and value (V) weights. Our momentum-based filtering mechanism is applied to each partition individually. We also add the explanations in Appendix D.4 of the revised manuscript.
>
> ---
> ## **Question 4**
> *In the experiment setting, is it fair to directly compare LoRA with the fully fine-tuned MoFO? It looks like a more fair comparison is to set them under the same trainable parameters set, like comparing LoRA with MoFO(PEFT version), and comparing Default FT with MoFO(FT version).*
>
> ## **Response to Question 4:**
>
> Thanks for your comment.
>
> In the fine-tuning stage, the trainable parameter space of LoRA is a low-rank subspace. We understand that your proposed comparison of LoRA and MoFO(PEFT version) essentially involves evaluating MoFO and Adam within the specific trainable parameter space defined by LoRA. To explore this, we conduct comparison experiment following the setting in Table 1 of Section 4 in the revised manuscript. The table below indicates that MoFO + LoRA mitigates forgetting in LoRA.
>
> | Method                     | GSM8K (FT) |       | Avg. of General Capacities |       | CR (General)  | MMLU (General) | Coding (General) |
> |----------------------------|------------|-------|----------------------------|-------|---------------|----------------|------------------|
> | LoRA                       | **43.3**   |       | 43.1                       |       | 65.1          | 37.7           | 26.4             |
> | LoRA + MoFO($\alpha$%=50%) | 42.1       |       | **43.9**                   |       | 65.5      | 39.4       | 26.7         |

---

> ### Author Response · Authors · 2024-12-01
> **Seeking for your Valuable Feedback**
>
> Dear Reviewer,
>
> Thank you for the time and effort you have dedicated to reviewing our paper. We sincerely appreciate your insightful comments and suggestions. We have conducted additional experiments and analyses to address your concerns and questions in the responses above. Please feel free to review them at your convenience.
>
> Your comments have been highly valuable to us, and we have put much effort into running new experiments to address your concern. We would love to receive feedback from you. If your concern is addressed, we humbly invite the reviewer to consider increasing the score. Your support is deeply appreciated!

---

> > ### Comment · Reviewer_xNNU · 2024-12-02
> >
> > Sorry for the delay. The authors' responses have addressed most of my concerns. I will increase my evaluation accordingly. Thanks!

---

### Official Review · Reviewer_sDCx · 2024-11-04

**Soundness:** 3
**Presentation:** 4
**Contribution:** 3
**Rating:** 6
**Confidence:** 4

**Summary:**

The authors focus on mitigating forgetting of pre-training knowledge in LLM fine-tuning. They propose update models with only a subset of parameters based on momentum during fine-tuning. Experiments on public datasets are conducted to demonstrate the effectiveness of the proposed method.

**Strengths:**

1. The paper follows a clear, well-thought-out structure that facilitates easy reading and comprehension;

2. The experimental results appear encouraging.

**Weaknesses:**

1. In Section 2.1, the authors show a strong correlation between forgetting and the distance from the pre-trained model with experiments. However, no analysis or proof is provided to explain the reason.

2. Considering that MoFO only uses a subset of parameters during fine-tuning, it may require more time or iterations compared to other methods to reach a minimum loss. It could be helpful to provide a time cost analysis section.

3. Clarification on the selection process for $\alpha$ would enhance the understanding of this parameter's role.

4. Table 5 demonstrates that MoFO outperforms Gradient-filtered BCD and MV-filtered BCD. A deeper analysis of the factors contributing to MoFO's superior performance is needed to provide further insight into these results.

5. In Fig. 4, the differing numbers of points for various methods and the significance of lighter colors could be further clarified for better interpretability.

**Questions:**

N/A

---

> ### Author Response · Authors · 2024-11-25
> **Response (Part I)**
>
> ## **Weakness 1**
> *In Section 2.1, the authors show a strong correlation between forgetting and the distance from the pre-trained model with experiments. However, no analysis or proof is provided to explain the reason.*
>
> ## **Response to Weakness 1:**
>
> We conduct some initial theoretical analysis on why MoFO outperforms Adam in mitigating forgetting. We extend the illustrating example in Section 5 of our revised manuscript (Section 4 in the original manuscript) to a higher-dimensional case. Since MoFO’s filtering mechanism is to greedily update the parameter with the largest momentum magnitude, we simplify the comparison between Adam and MoFO to a comparison between **vanilla gradient descent (vanilla GD) and greedy BCD**. That is, we ignore the impact of momentum for simplicity.
>
> ### **Problem Setup**
> Suppose the parameter space is $\mathbb{R}^d$, and the updating ratio is $\alpha$% = $1/d$, i.e., only one coordinate is updated in each iteration. We assume the pre-training loss is $L_{pretrain}(\theta) = \frac{1}{2} \Vert \theta \Vert_2^2,$ and that the model has been trained to the global minimum $\theta_{\rm pretrain} = (0, 0, \dots, 0)$ during the pre-training phase. The fine-tuning loss is given by
> $\mathcal{L}(\theta) = \prod_{i=1}^d (a_i \theta_i - b_i)^2,$
> where $a_i, b_i > 0$ for any $1 \leq i \leq d$. So the set of global minima of $\mathcal{L}(\theta)$ is
> $$
> S = \bigcup_{i=1}^d S_i, \quad S_i := \{\theta \in \mathbb{R}^d: \theta_i = b_i / a_i\}.
> $$
> Each $S_i$ is a hyperplane.
>
> **Remarks:**
> We note that the loss landscape of neural networks are highly non-convex [1]. Here, we also adopt a non-convex fine-tuning loss $\mathcal{L}(\theta)$. Further, we note that the set of global-minima $S$ consists of infinitely many minima spread across multiple hyperplanes, aligning with the observation on the degenerate structure of minima in neural networks [2].
>
> ### **Theoretical Analysis**
>
> **Theorem 1.** In this example, with appropriate learning rates, greedy BCD converges to a minimum closer to the pre-training state compared to vanilla GD. Consequently, greedy BCD preserves a lower pre-training loss.
>
> **Proof of Theorem 1:** Let the set $U := \{\theta \in \mathbb{R}^d : \theta_i < b_i / a_i, \ \forall 1 \leq i \leq d\}$. We note that:
> 1. The boundary of $U$ is the subset of $S = \cup_{i=1}^d S_i$, which is the collection all global minima of the fine-tuning loss.
> 2. The pre-training state $\theta_{\rm pretrain} = (0, 0, \dots, 0)$, which is also the starting point of fine-tuning, lies in $U$.
>
> We may as well assume that with proper learning rates, the parameter $\theta$ remains within $U$ during training, unless it converges to a minimum on the boundary. If it goes across the boundary at a certain iteration before converging, the learning rate can be adjusted to ensure that it remains within $U$.
>
> **Analysis of Greedy BCD**
> Taking the derivative of $\mathcal{L}(\theta)$ with respect to $\theta_i$,
> $$
> \frac{\partial \mathcal{L}}{\partial \theta_i} = 2a_i(a_i \theta_i - b_i) \prod_{j \neq i} (a_j \theta_j - b_j)^2 = \frac{2 \mathcal{L}(\theta)}{\theta_i - \frac{b_i}{a_i}}.
> $$
> At any point $\theta \in U$, greedy BCD selects the coordinate
> $$
> i_0 \in \arg \max_{1 \leq i \leq d} \bigg| \frac{\partial \mathcal{L}}{\partial \theta_i} \bigg| = \arg \max_{1 \leq i \leq d}  \frac{1}{\frac{b_i}{a_i} - \theta_i}  = \arg \min_{1 \leq i \leq d} (\frac{b_i}{a_i} - \theta_i).
> $$
>
> We will prove that greedy BCD converges along the same coordinate to the minimum. For clarity of definition, we let $\theta_t$ represent the parameters at iteration $t$, with $\theta_{i, t}$ denoting the $i$-th coordinate.
>
> Suppose we are at a fixed iteration $t$, where $i_0 \in \arg \min_{1 \leq i \leq d} \left\\{ \frac{b_i}{a_i} - \theta_i \right\\}$, and greedy BCD selects coordinate $i_0$ as the update direction. The parameter updates are:
> $$
> \theta_{i_0, t+1} = \theta_{i_0, t} + \frac{2 \eta_t \mathcal{L}(\theta_t)}{\frac{b_i}{a_i} - \theta_{i_0, t}} > \theta_{i_0, t}; \quad \theta_{i, t+1} = \theta_{i, t}, \ \forall i \neq i_0.
> $$
>
> If the algorithm has not converged at iteration step $t+1$, then we have $\theta_{i_0, t} < b_{i_0} / a_{i_0}$. Moreover, for any $i \neq i_0$,
> $$
> \frac{b_i}{a_i} - \theta_{i_0, t+1} < \frac{b_i}{a_i} - \theta_{i_0, t} \leq \frac{b_i}{a_i} - \theta_{i, t} = \frac{b_i}{a_i} - \theta_{i, t+1}.
> $$
>
> Thus, at iteration $t+1$, $i_0$ becomes the only coordinate in $\arg \max_{1 \leq i \leq d} \left| \frac{\partial \mathcal{L}}{\partial \theta_i} \right|$, and greedy BCD will still update in the direction of coordinate $i_0$.
>
> Greedy BCD consistently updates $\theta_{i_0}$, eventually converging to $\theta^*_{\text{greedy}} = (0, \dots, 0, \frac{b_{i_0}}{a_{i_0}}, 0, \dots, 0)$, with pre-training loss
> $$
> \mathcal{L}_ {pretrain}(\theta^*_{\text{greedy}}) = \frac{b_{i_0}^2}{2a_{i_0}^2}.
> $$

---

> ### Author Response · Authors · 2024-11-25
> **Response (Part II) (Theorem 1 proof Cont'd)**
>
> **Analysis of Vanilla GD**
>
> For any $1 \leq i \leq d$, GD updates each coordinate as
> $$
> \theta_{i, t+1} = \theta_{i, t} + \frac{2 \eta_t \mathcal{L}(\theta_t)}{\theta_{i, t} - \frac{b_i}{a_i}} > \theta_{i, t}.
> $$
> So unlike greedy BCD, vanilla GD updates all the parameters. Assuming that the vanilla GD converges to $\theta^*_{\text{GD}}$, we have
> - $\theta^*_{\text{GD}, i} > 0$ for any $1 \leq i \leq d$,
> - There exists $j_0$ such that $\theta^*_{\text{GD}, j_0} = b_{j_0} / a_{j_0}$.
>
> Recall that at iteration 0, greedy BCD selects
> $$
> i_0 \in \arg \min_{1 \leq i \leq d} \bigg\\{\frac{b_i}{a_i} - \theta_{i, {\rm pretrain}}\bigg\\} = \arg \min_{1 \leq i \leq d} \bigg\\{\frac{b_i}{a_i}\bigg\\}.
> $$
>
> Thus, the pre-training loss for vanilla GD is
> $$
> \mathcal{L}_ {\rm pretrain}(\theta^*_ {\text{GD}}) = \frac{b_ {j_ 0}^2}{2a_ {j_ 0}^2} + \sum_ {i \neq j_ 0} \theta^{*2}_ {\text{GD}, i} > \frac{b_ {j_ 0}^2}{2a_ {j_ 0}^2} \geq \frac{b_ {i_ 0}^2}{2a_ {i_ 0}^2} =  \mathcal{L}_ {\rm pretrain}(\theta_ {greedy}).
> $$
>
> In conclusion, greedy BCD converges to a minimum closer to the pre-training state than GD, preserving a lower pre-training loss.
>
> We believe that the initial theoretical analysis on this example provides insights into the effectiveness of the filtering mechanism in mitigating forgetting. A more general analysis of MoFO and Adam is left for future work.
>
> **References:**
>
> [1] Liu, Chaoyue, Libin Zhu, and Mikhail Belkin. "Loss landscapes and optimization in over-parameterized non-linear systems and neural networks." Applied and Computational Harmonic Analysis 59 (2022): 85-116.
>
> [2] Lin, Zhanran, Puheng Li, and Lei Wu. "Exploring Neural Network Landscapes: Star-Shaped and Geodesic Connectivity." arXiv preprint arXiv:2404.06391 (2024).

---

> ### Author Response · Authors · 2024-11-25
> **Response (Part III)**
>
> ## **Weakness 2**
> *Considering that MoFO only uses a subset of parameters during fine-tuning, it may require more time or iterations compared to other methods to reach a minimum loss. It could be helpful to provide a time cost analysis section.*
>
> ## **Response to Weakness 2**
>
> We appreciate your suggestion. In our implementation, MoFO and the default FT use **the same number of iterations**. But they **achieve comparable fine-tuning performance and the additional time required by MoFO is minor** due to MoFO's efficient selection of parameter subsets. We provide an efficiency analysis as follows.
>
> (a). **Fine-tuning process:** We fine-tune the LLaMA2-7B model on the MetaMathQA dataset and evaluate the fine-tuning performance on the GSM8K benchmark during the training process. The results are presented in Figure 9 in Appendix E.4 of our revised manuscript. MoFO and default FT achieve comparable performance across the whole training process, **demonstrating that updating only a subset of parameters within each iteration in MoFO does not adversely affect convergence speed or final performance**.
>
> (b). **Total training time:** Although MoFO requires computing a mask to select parameters with the largest momentum magnitudes, the additional computational overhead is minimal. Following the setting in Section 4.2 in the revised manuscript, we measured the training times for MoFO and default FT on different model sizes and datasets. The results are as follows:
>
> | Model                        | Default FT    | MoFO ($\alpha$% = 10%)             | Additional Training Time    |
> |------------------------------|-----------------------------|---------------------------|----------|
> | LLaMA3.2-1B                  | 49m22s       | 51m24s     | 4.1% |
> | LLaMA3.2-3B                  | 1h30m18s| 1h34m24s | 4.5% |
> | LLaMA3-8B on the UltraFeedback dataset[3] | 5h2m0.69s | 5h17m34.01s | 5.0%  |
>
> As shown in the table, the additional training time incurred by MoFO is approximately 4–5%, which is relatively minor and manageable in practical applications.
>
> **Reference:**
>
> [3] Cui, Ganqu, et al. "ULTRAFEEDBACK: Boosting Language Models with Scaled AI Feedback." Forty-first International Conference on Machine Learning. 2024.
>
> ---
> ## **Weakness 3**
> *Clarification on the selection process for would enhance the understanding of this parameter's role.*
>
>
> ## **Response to Weakness 3:**
>
> We select the fraction $\alpha$% through grid search. The principle of choosing $\alpha$% is to find the optimal balance where LLMs perform well on fine-tuning tasks without significant forgetting of pre-trained knowledge.
>
> Moreover, we find that the fraction hyperparameter $\alpha$% affects models of different sizes in a consistent manner; its optimal value remains approximately the same regardless of model size. To explore this, we fine-tune LLMs of various sizes (1B, 3B, 7B) with different fraction ratios ($\alpha$% = 5%, 10%, 20%, 40%, 80%). The results of this experiment are presented in Figure 6 and Figure 7 of Appendix E.1 in our revised manuscript, and we make the following key observations.
>
> **Key Observations:**
>
> a. **Good Fine-Tuning Performance at $\alpha$% $\approx$ 20%:** Across models of different sizes, setting the fraction $\alpha$% to approximately 20% allows MoFO to reach fine-tuning performance similar to the default FT (with up to 3% performance drop). Therefore, we recommend initiating with a 20\% update fraction when applying MoFO on new LLMs or fine-tuning tasks.
>
> b. **Enhanced Knowledge Retention in Larger Models:** Larger models tend to better retain the pre-training knowledge with MoFO. Specifically, for a given fraction ratio, MoFO provides more significant performance gains in mitigating forgetting as the model grows larger.

---

> ### Author Response · Authors · 2024-11-25
> **Response (Part IV)**
>
> ## **Weakness 4**
> *Table 5 demonstrates that MoFO outperforms Gradient-filtered BCD and MV-filtered BCD. A deeper analysis of the factors contributing to MoFO's superior performance is needed to provide further insight into these results.*
>
> ## **Response to Weakness 4:**
> Thank you for your insightful comment. **We hypothesize that the benefit of momentum-filtered update mechanism leads to more stable and consistent updates during the training.** Specifically, we hypothesize the following:
>
> **(A). Utilizing momentum instead of gradient filtering leads to more stable updates.** Momentum accumulates historical gradients, so it promotes stability by smoothing out fluctuations in the gradient updates. Thereby, during the training process, the momentum-filtering mechanism chooses updating parameters in a more stable manner.
>
> **(B). Excluding the introduction of $v$ in the filtering mechanism contributes to more stable updates.** The 2nd-order moment $v_t$ may normalize gradients based on their magnitudes, potentially averaging out the importance of individual parameters within the filtering mechanism. Thereby, during the training process, not incorporating $v$ may help choose updating parameters in a more stable manner.
>
> For the ablation study, we add the GV-filtered BCD methods as a baseline, which replaces MoFO's filter by $\texttt{FLT}_{\alpha}(g_t / \sqrt{v_t})$. Following the setting of experiments in Figure 1, we run all four methods with the updating fraction $\alpha% = 3% over approximately 200 steps. To assess how many parameters change significantly during training, we calculate the percentage of weight parameters whose absolute change exceeds a threshold of 2e-6.
>
> | Method        | Percentage of Significant Parameter Updates |
> |---------------|---------------------------------------------|
> | MoFO          | **29.8%**                                       |
> | Gradient BCD  | 35.7%                                       |
> | MV-filtered BCD | 83.6%                                     |
> | GV-filtered BCD | 87.1%                                    |
>
> These results indicate that the parameter updating process of MoFO is more stable, which we think may contribute to its better fine-tuning performance.
>
> ---
> ## **Weakness 5**
>
> *In Fig. 4, the differing numbers of points for various methods and the significance of lighter colors could be further clarified for better interpretability.*
>
> ## **Response to Weakness 5:**
> We apologize for the confusion regarding Figure 4. We evaluate each method using a comparable number of hyperparameters. The differing numbers of solid points for various methods are due to the number of hyperparameter configurations that result in Pareto-optimal models for each method. For each method, we conduct experiments using a series of hyperparameter settings. All the evaluated models are plotted in the figure, but only those that meet the Pareto optimality criterion are represented as solid points on the Pareto front.
>
> The lighter, semi-transparent points represent the other evaluated models that do not lie on the Pareto front.
>
>
> We add the clarification on these points to enhance interpretability in Section 4.2 of our revised manuscript.

---

> ### Author Response · Authors · 2024-12-01
> **Seeking for your Valuable Feedback**
>
> Dear Reviewer,
>
> Thank you for the time and effort you have dedicated to reviewing our paper. We sincerely appreciate your insightful comments and suggestions. We have conducted additional experiments and analyses to address your concerns and questions in the responses above. Please feel free to review them at your convenience.
>
> Your comments have been highly valuable to us, and we have put much effort into running new experiments to address your concern. We would love to receive feedback from you. If your concern is addressed, we humbly invite the reviewer to consider increasing the score. Your support is deeply appreciated!

---

> > ### Comment · Reviewer_sDCx · 2024-12-03
> >
> > Thank you for your responses. They have addressed most of my concerns, and I have adjusted my score accordingly.

---

### Official Review · Reviewer_Bhr4 · 2024-11-04

**Soundness:** 3
**Presentation:** 3
**Contribution:** 3
**Rating:** 6
**Confidence:** 4

**Summary:**

This paper proposes a new method, named Momentum-Filtered Optimizer (MoFO) for finetuning LLMs while mitigating the forgetting of the knowledge captured by the pre-trained model. The proposed method leverages block coordinate descent (BCD), where the blocks of parameters to update are chosen based on the top-\alpha largest momentum values. The convergence result of MoFO is discussed. Empirical results are also provided to justify the advantages of MoFO.

**Strengths:**

1. The paper provides a simple but elegant approach to mitigating forgetting in LLM fine-tuning.

2. Empirical results are convincing.

3. The paper is well-written with illustrative figures.

**Weaknesses:**

1. Using the default partitioning of model parameters as implemented in PyTorch is quite an ad-hoc choice. Is there any better or more principled way to partition the model parameters?

2. The convergence result is good, but it does not show the advantages of MoFO over the traditional Adam method, especially in mitigating forgetting.

**Questions:**

1. In Table 1, why LoRA perform well on HumanEval but worse on other tasks? Also, in both Table 1 and Figure 4, the performance of MoFO is not significantly better than HFT.

2. The study of the MoFO approach applied on other optimization methods beyond Adam is recommended.

3. I would suggest the authors include  an efficiency analysis for MoFO.

**Details Of Ethics Concerns:**

I have no ethics concerns about the paper.

---

> ### Author Response · Authors · 2024-11-25
> **Response (Part I)**
>
> ## **Weakness 1**
>
> *Using the default partitioning of model parameters as implemented in PyTorch is quite an ad-hoc choice. Is there any better or more principled way to partition the model parameters?*
>
>
> ## **Response to Weakness 1:**
>
> Thanks very much for your valuable advice. Exploring better and more principled ways to partition the model parameters is indeed a meaningful and worthwhile direction, and we have made preliminary attempts in this area.
>
> In our work, we use the **default partitioning** scheme in PyTorch's Transformer implementation. Different types of parameters within the Transformer, such as query (Q), key (K), value (V) weights for attention heads, and feed-forward network (FFN) weights, are divided into separate partitions. Notably, in the default PyTorch implementation, within a layer, the query (Q) weights of all attention heads are grouped into a single partition. The same applies to the key (K) and value (V) weights. Our momentum-based filtering mechanism is applied to each partition individually.
>
> In **our alternative approach**, we partition the model parameters at the level of individual attention heads. Recent works [1][2] find that the Hessian matrix of transformers exhibits a near-block-diagonal structure with several dense principal sub-blocks. Specifically, [2] observes that within the same layer, different Q (or K) heads correspond to different blocks in the Hessian. Inspired by this observation, we modify our approach by treating the Q, K, and V weights from different attention heads as separate partitions. Our momentum-based filtering mechanism is then applied individually to these finer-grained partitions.
>
> Following the setting in Table 1 of Section 4, we conduct experiments on LLaMA3.2-1B and LLaMA2-7B models to compare the default partitioning with the alternative partitioning.
>
> **Table 1: LLaMA3.2-1B**
>
> | Method                                                         | GSM8K (FT) |       | Avg. of General Capacities |       | CR (General) | MMLU (General) | HumanEval (General) |
> |----------------------------------------------------------------|------------|-------|----------------------------|-------|--------------|----------------|---------------------|
> | MoFO with default partition ($\alpha$% = 10%)               | **27.1**   |       | 39.6                       |       | 52.6         | 36.4       | 29.8                |
> | MOFO with individual head partition ($\alpha$% = 10%)       | 27.0       |       | **40.1**                   |       | 52.8     | 36.3           | 31.2            |
>
> **Table 2: Llama-2-7B**
>
> | Method                                                    | GSM8K (FT) |       | Avg. of General Capacities |       | CR (General) | MMLU (General) | HumanEval (General) |
> |-----------------------------------------------------------|------------|-------|----------------------------|-------|--------------|----------------|---------------------|
> | MoFO with default partition ($\alpha$%=15%)          | **47.7**   |       | **44.3**                   |       | 65.7     | 42.7       | 24.6            |
> | MoFO with individual head partition ($\alpha$%=15%)  | 46.6       |       | 43.8                       |       | 65.7     | 41.6           | 24.1                |
>
>
> The results shows both partition schemes yield similar performance. For the smaller model LLaMA3.2-1B, individual head partition leads to a slightly better forgetting mitigation performance.
>
> From the above initial experiment results, we believe that there is room for further exploration in better partitioning strategies.
>
> **References**
>
> [1] Zhang, Yushun, et al. "Adam-mini: Use fewer learning rates to gain more." *arXiv preprint arXiv:2406.16793* (2024).
>
> [2] Zhang, Yushun, et al. "Why Transformers Need Adam: A Hessian Perspective." *arXiv preprint arXiv:2402.16788* (2024).

---

> ### Author Response · Authors · 2024-11-25
> **Response (Part II)**
>
> ## **Weakness 2**
> *The convergence result is good, but it does not show the advantages of MoFO over the traditional Adam method, especially in mitigating forgetting.*
>
>
> ## **Response to Weakness 2:**
>
> We conduct some initial theoretical analysis on why MoFO outperforms Adam in mitigating forgetting. We extend the illustrating example in Section 5 of our revised manuscript (Section 4 in the original manuscript) to a higher-dimensional case. Since MoFO’s filtering mechanism is to greedily update the parameter with the largest momentum magnitude, we simplify the comparison between Adam and MoFO to a comparison between **vanilla gradient descent (vanilla GD) and greedy BCD**. That is, we ignore the impact of momentum for simplicity.
>
> ### **Problem Setup**
> Suppose the parameter space is $\mathbb{R}^d$, and the updating ratio is $\alpha$% = $1/d$, i.e., only one coordinate is updated in each iteration. We assume the pre-training loss is $L_{pretrain}(\theta) = \frac{1}{2} \Vert \theta \Vert_2^2,$ and that the model has been trained to the global minimum $\theta_{\rm pretrain} = (0, 0, \dots, 0)$ during the pre-training phase. The fine-tuning loss is given by
> $\mathcal{L}(\theta) = \prod_{i=1}^d (a_i \theta_i - b_i)^2,$
> where $a_i, b_i > 0$ for any $1 \leq i \leq d$. So the set of global minima of $\mathcal{L}(\theta)$ is
> $$
> S = \bigcup_{i=1}^d S_i, \quad S_i := \{\theta \in \mathbb{R}^d: \theta_i = b_i / a_i\}.
> $$
> Each $S_i$ is a hyperplane.
>
> **Remarks:**
> We note that the loss landscape of neural networks are highly non-convex [3]. Here, we also adopt a non-convex fine-tuning loss $\mathcal{L}(\theta)$. Further, we note that the set of global-minima $S$ consists of infinitely many minima spread across multiple hyperplanes, aligning with the observation on the degenerate structure of minima in neural networks [4].
>
> ### **Theoretical Analysis**
>
> **Theorem 1.** In this example, with appropriate learning rates, greedy BCD converges to a minimum closer to the pre-training state compared to vanilla GD. Consequently, greedy BCD preserves a lower pre-training loss.
>
> **Proof of Theorem 1:** Let the set $U := \{\theta \in \mathbb{R}^d : \theta_i < b_i / a_i, \ \forall 1 \leq i \leq d\}$. We note that:
> 1. The boundary of $U$ is the subset of $S = \cup_{i=1}^d S_i$, which is the collection all global minima of the fine-tuning loss.
> 2. The pre-training state $\theta_{\rm pretrain} = (0, 0, \dots, 0)$, which is also the starting point of fine-tuning, lies in $U$.
>
> We may as well assume that with proper learning rates, the parameter $\theta$ remains within $U$ during training, unless it converges to a minimum on the boundary. If it goes across the boundary at a certain iteration before converging, the learning rate can be adjusted to ensure that it remains within $U$.
>
> **Analysis of Greedy BCD**
> Taking the derivative of $\mathcal{L}(\theta)$ with respect to $\theta_i$,
> $$
> \frac{\partial \mathcal{L}}{\partial \theta_i} = 2a_i(a_i \theta_i - b_i) \prod_{j \neq i} (a_j \theta_j - b_j)^2 = \frac{2 \mathcal{L}(\theta)}{\theta_i - \frac{b_i}{a_i}}.
> $$
> At any point $\theta \in U$, greedy BCD selects the coordinate
> $$
> i_0 \in \arg \max_{1 \leq i \leq d} \bigg| \frac{\partial \mathcal{L}}{\partial \theta_i} \bigg| = \arg \max_{1 \leq i \leq d}  \frac{1}{\frac{b_i}{a_i} - \theta_i}  = \arg \min_{1 \leq i \leq d} (\frac{b_i}{a_i} - \theta_i).
> $$
>
> We will prove that greedy BCD converges along the same coordinate to the minimum. For clarity of definition, we let $\theta_t$ represent the parameters at iteration $t$, with $\theta_{i, t}$ denoting the $i$-th coordinate.
>
> Suppose we are at a fixed iteration $t$, where $i_0 \in \arg \min_{1 \leq i \leq d} \left\\{ \frac{b_i}{a_i} - \theta_i \right\\}$, and greedy BCD selects coordinate $i_0$ as the update direction. The parameter updates are:
> $$
> \theta_{i_0, t+1} = \theta_{i_0, t} + \frac{2 \eta_t \mathcal{L}(\theta_t)}{\frac{b_i}{a_i} - \theta_{i_0, t}} > \theta_{i_0, t}; \quad \theta_{i, t+1} = \theta_{i, t}, \ \forall i \neq i_0.
> $$
>
> If the algorithm has not converged at iteration step $t+1$, then we have $\theta_{i_0, t} < b_{i_0} / a_{i_0}$. Moreover, for any $i \neq i_0$,
> $$
> \frac{b_i}{a_i} - \theta_{i_0, t+1} < \frac{b_i}{a_i} - \theta_{i_0, t} \leq \frac{b_i}{a_i} - \theta_{i, t} = \frac{b_i}{a_i} - \theta_{i, t+1}.
> $$
>
> Thus, at iteration $t+1$, $i_0$ becomes the only coordinate in $\arg \max_{1 \leq i \leq d} \left| \frac{\partial \mathcal{L}}{\partial \theta_i} \right|$, and greedy BCD will still update in the direction of coordinate $i_0$.
>
> Greedy BCD consistently updates $\theta_{i_0}$, eventually converging to $\theta^*_{\text{greedy}} = (0, \dots, 0, \frac{b_{i_0}}{a_{i_0}}, 0, \dots, 0)$, with pre-training loss
> $$
> \mathcal{L}_ {pretrain}(\theta^*_{\text{greedy}}) = \frac{b_{i_0}^2}{2a_{i_0}^2}.
> $$

---

> ### Author Response · Authors · 2024-11-25
> **Response (Part III) (Theorem 1 proof Cont'd)**
>
> **Analysis of Vanilla GD**
>
> For any $1 \leq i \leq d$, GD updates each coordinate as
> $$
> \theta_{i, t+1} = \theta_{i, t} + \frac{2 \eta_t \mathcal{L}(\theta_t)}{\theta_{i, t} - \frac{b_i}{a_i}} > \theta_{i, t}.
> $$
> So unlike greedy BCD, vanilla GD updates all the parameters. Assuming that the vanilla GD converges to $\theta^*_{\text{GD}}$, we have
> - $\theta^*_{\text{GD}, i} > 0$ for any $1 \leq i \leq d$,
> - There exists $j_0$ such that $\theta^*_{\text{GD}, j_0} = b_{j_0} / a_{j_0}$.
>
> Recall that at iteration 0, greedy BCD selects
> $$
> i_0 \in \arg \min_{1 \leq i \leq d} \bigg\\{\frac{b_i}{a_i} - \theta_{i, {\rm pretrain}}\bigg\\} = \arg \min_{1 \leq i \leq d} \bigg\\{\frac{b_i}{a_i}\bigg\\}.
> $$
>
> Thus, the pre-training loss for vanilla GD is
> $$
> \mathcal{L}_ {\rm pretrain}(\theta^*_ {\text{GD}}) = \frac{b_ {j_ 0}^2}{2a_ {j_ 0}^2} + \sum_ {i \neq j_ 0} \theta^{*2}_ {\text{GD}, i} > \frac{b_ {j_ 0}^2}{2a_ {j_ 0}^2} \geq \frac{b_ {i_ 0}^2}{2a_ {i_ 0}^2} =  \mathcal{L}_ {\rm pretrain}(\theta_ {greedy}).
> $$
>
> In conclusion, greedy BCD converges to a minimum closer to the pre-training state than GD, preserving a lower pre-training loss.
>
> We believe that the initial theoretical analysis on this example provides insights into the effectiveness of the filtering mechanism in mitigating forgetting. A more general analysis of MoFO and Adam is left for future work.
>
> **References:**
>
> [3] Liu, Chaoyue, Libin Zhu, and Mikhail Belkin. "Loss landscapes and optimization in over-parameterized non-linear systems and neural networks." Applied and Computational Harmonic Analysis 59 (2022): 85-116.
>
> [4] Lin, Zhanran, Puheng Li, and Lei Wu. "Exploring Neural Network Landscapes: Star-Shaped and Geodesic Connectivity." arXiv preprint arXiv:2404.06391 (2024).

---

> ### Author Response · Authors · 2024-11-25
> **Response (Part IV)**
>
> ## **Questions 1**
> *In Table 1, why LoRA perform well on HumanEval but worse on other tasks? Also, in both Table 1 and Figure 4, the performance of MoFO is not significantly better than HFT.*
>
> ## **Response to Question 1:**
>
> Thanks for your thoughtful questions. Our responses are as follows:
>
> **(a).** The observation on LoRA is noteworthy. In all our experiments (Table 1, Table 5, Table 6 and Table 7 in the revised manuscript), LoRA exhibit strong performance in mitigating forgetting of **coding abilities** across various tasks. So far, we are not fully awared of the reason of this phenomenon, and it deserves further investigation. We conjecture that this is due to certain implicit bias of LoRA. We add a remark to highlight this point in the Section 4.2 in the revised manuscript.
>
> **(b).** In Table 1 of our manuscript, both MoFO and HFT achieve fine-tuning performance comparable to the **default FT**, and maintain consistency in general capability scores of the **pre-trained model**. This indicates that, in this setting, both methods attain **nearly optimal performance in forgetting mitigation**. However, in other tasks (e.g., Tables 5 and 6 of our manuscript), MoFO demonstrates significantly superior general capabilities compared to HFT while maintaining similar fine-tuning performance.
>
> ---
> ## **Question 2**
> *The study of the MoFO approach applied on other optimization methods beyond Adam is recommended.*
>
> ## **Response to Question 2:**
>
> Thank you for the recommendation. We apply MoFO to another optimizer, Lion [5], by utilizing the same momentum-filtering mechanism. To be more specific, at iteration $t$, the Lion optimizer follows the algorithm:
> $$
> g_t \leftarrow \nabla_\theta f\left(\theta_{t-1}\right)
> $$
> $$
> c_t \leftarrow \beta_1 m_{t-1}+\left(1-\beta_1\right) g_t
> $$
> $$
> \theta_t \leftarrow \theta_{t-1}-\eta_t\left(\operatorname{sign}\left(c_t\right)+\lambda \theta_{t-1}\right),
> $$
> where the momentum is updated as follows:
> $$
> m_t \leftarrow \beta_2 m_{t-1}+\left(1-\beta_2\right) g_t.
> $$
>
> Integrating MoFO into Lion involves applying a filter to the momentum term before updating the parameter $\theta$, specifically:
> $$
> g_t \leftarrow \nabla_\theta f\left(\theta_{t-1}\right)
> $$
> $$
> c_t \leftarrow \beta_1 m_{t-1}+\left(1-\beta_1\right) g_t
> $$
> $$
> \theta_t \leftarrow \theta_{t-1}-\eta_t \left(\operatorname{sign}\left(c_t \odot FLT_{\alpha}(m_{t-1})\right) +\lambda \theta_{t-1}\right).
> $$
>
> We follow the experimental setup described in Figure 1 of our manuscript and conduct additional experiments applying MoFO to the Lion optimizer. The experimental results demonstrate that integrating MoFO with Lion significantly alleviates forgetting while achieving comparable loss values to the standard Lion optimizer, though the results are still worse than those obtained by MoFO for Adam.
>
> |                | HellaSwag | ARC-easy | ARC-challenge | Average |
> |----------------|-----------|----------|---------------|---------|
> | **Pythia-160m** | 30.1      | 39.6     | 23.8          | 31.2    |
> | **Adam**        | 28.3      | 37.4     | 22.1          | 29.3    |
> | **MoFO (for Adam)**        | 29.9      | 42.0     | 22.9          | 31.6    |
> | **Lion**        | 26.5      | 29.0     | 24.1          | 26.5    |
> | **Lion + MoFO** | 27.5      | 36.7     | 23.2          | 29.1    |
>
> **Reference:**
>
> [5] Chen, Xiangning, et al. "Symbolic discovery of optimization algorithms." Advances in neural information processing systems 36 (2024).

---

> ### Author Response · Authors · 2024-11-25
> **Response (Part V)**
>
> ## **Question 3**
> *I would suggest the authors include an efficiency analysis for MoFO.*
>
> ## **Response to Question 3:**
>
> Thanks for your advice. We claim that MoFO does not lead to significant reduced fine-tuning efficiency. We provide an efficiency analysis by comparing the fine-tuning process and total training time between MoFO and default FT as follows.
>
> (a). **Fine-tuning process:** We fine-tune the LLaMA2-7B model on the MetaMathQA dataset and evaluate the fine-tuning performance on the GSM8K benchmark during the training process. The results are presented in Figure 9 in Appendix E.4 of our revised manuscript. MoFO and default FT achieve comparable performance across the whole training process, demonstrating that **updating only a subset of parameters within each iteration in MoFO does not adversely affect convergence speed or final performance**.
>
> (b). **Total training time:** Although MoFO requires computing a mask to select parameters with the largest momentum magnitudes, the additional computational overhead is minor. Following the setting in Section 4.2 in the revised manuscript, we measured the training times for MoFO and default FT on different model sizes and datasets. The results are as follows:
>
> | Model                        | Default FT    | MoFO ($\alpha$% = 10%)             | Additional Training Time    |
> |------------------------------|-----------------------------|---------------------------|----------|
> | LLaMA3.2-1B                  | 49m22s       | 51m24s     | 4.1% |
> | LLaMA3.2-3B                  | 1h30m18s| 1h34m24s | 4.5% |
> | LLaMA3-8B on the UltraFeedback dataset[6] | 5h2m0.69s | 5h17m34.01s | 5.0%  |
>
>
> As shown in the table, the additional training time incurred by MoFO is approximately 4–5%, which is relatively minor and manageable in practical applications.
>
> **Reference**
>
> [6] Cui, Ganqu, et al. "ULTRAFEEDBACK: Boosting Language Models with Scaled AI Feedback." Forty-first International Conference on Machine Learning. 2024.

---

### Official Review · Reviewer_UETj · 2024-11-04

**Soundness:** 3
**Presentation:** 3
**Contribution:** 3
**Rating:** 6
**Confidence:** 3

**Summary:**

This paper presents a novel fine-tuning algorithm called the Momentum-Filtered Optimizer (MoFO), designed to address the issue of knowledge forgetting in large language models (LLMs) during fine-tuning. As a replay and regularization free strategy, MoFO iteratively selects and updates model parameters based on the largest momentum magnitudes, extending greedy block coordinate descent (BCD) methods. The algorithm achieves comparable fine-tuning performance to standard methods while effectively mitigating knowledge forgetting, without requiring access to pre-training data. The authors validate MoFO through rigorous convergence analysis and extensive experiments, demonstrating its superiority in reducing forgetting, making it particularly suitable for scenarios involving checkpoint-only open-source LLMs.

**Strengths:**

1. The paper deals with an important problem in LLM application and gives a practical solution with fair theoretical support.
2. The illustation and analysis of MoFO strategy is clear and in detail.
3. Authors supported their method with experiments covering various tasks and settings.

**Weaknesses:**

1. The discussion of this paper did not cover other algorithms that are non replay and non regularization, and therefore doesn't give enough support for the motivation and novelty of their work.
(including but not limited to: https://arxiv.org/pdf/2309.06256, https://arxiv.org/pdf/2404.10306, https://arxiv.org/abs/2302.03241 etc.)
A brief literature review section might be added specifically comparing MoFO to these other approaches, highlighting key differences and potential advantages.
2. The number and quality of baselines compared in experiments are too limited; the experimental validation of orthogonal methods is somehow unconvincing without proper comparisons.

**Questions:**

1. Add discussions and reference for more works (see list in Weakness 1) on LLM CF mitigation, especially justify your advantage compared to them
2. Add baselines (see list in Weakness 1) and comparison studies (i.e. is the combination of MoFO and replay better than both MoFO only and replay only?) correspondingly in experimental studies. You might also need to better verify (possibly by adding references) why the experimental success of the combination of MoFO with your choice of HFT, GEM and replay would be representative enough to suggest the effectiveness of combining MoFO with othorgonal strategies.

---

> ### Author Response · Authors · 2024-11-25
> **Response (Part I)**
>
> ### **Weaknesses 1**
> *The discussion of this paper did not cover other algorithms that are non replay and non regularization, and therefore doesn't give enough support for the motivation and novelty... A brief literature review section might be added specifically comparing MoFO to these other approaches, highlighting key differences and potential advantages.*
>
> ### **Response to Weakness 1:**
>
> Thank you for highlighting these relevant works. In our **revised manuscript**, we have added a "Related Work" section (Section 2), which enriches the references and outlines the key differences between MoFO and existing methods, as well as the potential advantages of MoFO.
>
> Specifically, the three papers [1-3] mentioned in the review comment are all included in this section. To provide a clearer framework for comparison, we have categorized forgetting-mitigation methods into five distinct types: replay-based, regularization-based, model merging, architecture-based, and optimization-based methods. Specifically, [1] is classified as a model-merging method, whereas [2], [3], and MoFO are categorized as optimization-based methods.
>
> For more comparative experiments among these methods, please refer to our response to Question 2.
>
> **References:**
>
> [1] Lin, Yong, et al. "Mitigating the alignment tax of rlhf." Proceedings of the 2024 Conference on Empirical Methods in Natural Language Processing. 2024.
>
> [2] Zhang, Hengyuan, et al. "Balancing speciality and versatility: a coarse to fine framework for supervised fine-tuning large language model." arXiv preprint arXiv:2404.10306 (2024).
>
> [3] Ke, Zixuan, et al. "Continual pre-training of language models." arXiv preprint arXiv:2302.03241 (2023).
>
> ---
> ### **Weakness 2**
> *The number and quality of baselines compared in experiments are too limited; the experimental validation of orthogonal methods is somehow unconvincing without proper comparisons.*
>
> ### **Response to Weakness 2:**
>
> We have added comparisions to more baselines in the experiments to address this comment. Please our response to Question 2 for details.
>
> ---
> ### **Question 1**
> *Add discussions and reference for more works (see list in Weakness 1) on LLM CF mitigation, especially justify your advantage compared to them.*
>
> ### **Response to Question 1:**
>
> Please see our response to Weakness 1.

---

> > ### Author Response · Authors · 2024-11-25
> > **Response (Part IV)**
> >
> > ### **Response to Question 2**
> >
> > **\(C). Comparision and orthogonal studies on continual learning**
> >
> > To present our results more clearly, we have re-organized our findings into **tables**.
> >
> > **Comparision studies**
> >
> > Table 2 indicates that MoFO, when used alone, outperforms other baseline methods expect Replay. It is reasonable because Replay use the past data and MoFO is a replay-free method. We will show in Table 3.3 that the combination of MoFO and Replay can outperform Replay alone, which indicating the effectiveness of MoFO. (In these tables, OP and BWT are two performance metrics of continual learning, standing for overall performance and BackWard Transfer, respectively. Higher OP and BWT values reflect better mitigation of forgetting.)
> >
> > **Table 2: Performance Comparison of MoFO in the continual learning experiments**
> >
> > | Method       | OP   | BWT   |
> > |--------------|------|-------|
> > | Default FT   | 38.4 | -10.3 |
> > | HFT          | 39.9 | -10.1 |
> > | MoFO ($\alpha$%=5%) | **41.3** | **-5.4**  |
> > | GEM          | 40.8 | -8.5  |
> > | EWC          | 41.1 | -8.3  |
> > | Replay               | **45.5** | **4.7**   |
> >
> > ---
> > **Orthogonal studies**
> >
> > Tables 3.1, 3.2, and 3.3 demonstrate that MoFO, when combined with other orthogonal methods, outperforms both the use of MoFO alone and the use of the orthogonal method alone. The only exception is the combination of MoFO and GEM, where MoFO+GEM outperforms GEM alone, but exhibits slightly lower performance in BWT than MoFO alone.
> >
> > **Table 3.1: Orthogonal Combination of MoFO with GEM**
> > | Method            | OP   | BWT   |
> > |-------------------|------|-------|
> > | GEM               | 40.8 | -8.5  |
> > | MoFO ($\alpha$%=5%) | 41.3 | **-5.4**  |
> > | GEM + MoFO        | **41.7** | -6.7  |
> >
> > **Table 3.2: Orthogonal Combination of MoFO with EWC**
> > | Method            | OP   | BWT   |
> > |-------------------|------|-------|
> > | EWC               | 41.1 | -8.3  |
> > | MoFO ($\alpha$%=5%) | 41.3 | -5.4  |
> > | EWC + MoFO        | **43.2** | **-4.4**  |
> >
> > **Table 3.3: Orthogonal Combination of MoFO with Replay**
> > | Method            | OP   | BWT   |
> > |-------------------|------|-------|
> > | Replay               | 45.5 | 4.7   |
> > | MoFO ($\alpha$%=5%) | 41.3 | -5.4  |
> > | Replay + MoFO        | **47.0** | **4.8**   |
> >
> > We believe these experimental results strengthen the validity of our experimental results.

---

> > > ### Comment · Reviewer_UETj · 2024-11-26
> > >
> > > Thanks for your responses and new experimental results, they have solved most of my concerns and I have modified some of my review scores accordingly.

---

> ### Author Response · Authors · 2024-11-25
> **Response (Part II)**
>
> ### **Question 2**
> *Add baselines (see list in Weakness 1) and comparison studies (i.e. is the combination of MoFO and replay better than both MoFO only and replay only?) correspondingly in experimental studies. You might also need to better verify (possibly by adding references) why the experimental success of the combination of MoFO with your choice of HFT, GEM and replay would be representative enough to suggest the effectiveness of combining MoFO with othorgonal strategies.*
>
> ### **Response to Question 2:**
>
> **(A). Comparision to the baselines [1][2][3] mentioned in Weakness 1**
>
> **Experiments on HMA [1]**
>
> Specifically, [1] proposes a heterogeneous model averaging (HMA) method between the pre-trained model and the fine-tuned model. Their approach evenly divides the LLM into three parts—the input part, the middle part, and the output part—and averages these parts with different ratios. In our model, LLaMA2-7B with 32 layers, this corresponds to layers 0-10 for the input part, layers 11-20 for the middle part, and layers 21-31 for the output part. _(Note: In the original paper, their model is OpenLLaMA with 26 layers, so the divisions are layers 1-8 for the input part, layers 9-17 for the middle part, and layers 18-26 for the output part.)_
>
>
> We select 15 different combinations of averaging ratios for different parts as follows: {(0.05, 0.2, 0.35), (0.1, 0.2, 0.3), (0.2, 0.2, 0.2), (0.3, 0.2, 0.1), (0.35, 0.2, 0.05), (0.3, 0.5, 0.7), (0.4, 0.5, 0.6), (0.5, 0.5, 0.5), (0.6, 0.5, 0.4), (0.7, 0.5, 0.3), (0.65, 0.8, 0.95), (0.7, 0.8, 0.9), (0.8, 0.8, 0.8), (0.9, 0.8, 0.7), (0.95, 0.8, 0.65)}.
>
>
> We employ the models obtained using the HMA method with these averaging ratios and plot their results to construct a Pareto front, as shown in Figure 10 in Appendix E.5. Our experimental results show that **the Pareto front of MoFO dominates that of the HMA methods, indicating that MoFO achieves better trade-offs between fine-tuning performance and knowledge retention**.
>
>
>
>
> **Experiments on CoFiTune [2] and SoftMasking [3]**
>
> Reference [2] introduces CoFiTune, a coarse-to-fine framework that balances specificity and versatility in LLMs by selectively updating specific modules and employing a soft-masking mechanism, which is introduced by [3]. We have compared MoFO with CoFiTune (with and without soft-masking) and the vanilla soft-masking method alone,  following the setting in Table 1 of Section 4. The results, presented in Table 1 below, demonstrate that MoFO outperforms these methods in both fine-tuning performance and mitigating forgetting. The results demonstrate that
> - CoFiTune achives similar forgetting mitigation performance as MoFO, but underperforms MoFO on fine-tuning tasks;
> - V-softmask exhibits slightly reduced performance in both fine-tuning tasks and mitigating forgetting than MoFO. These findings underscore the advantages of our proposed method.
>
> **Table 1: Comparison between MoFO, CoFiTune, and SoftMasking**
>
> | Method                | GSM8K (FT) |       | Avg. of General Capacities |       | CR (General) | MMLU (General) | HumanEval (General) |
> |-----------------------|------------|-------|----------------------------|-------|--------------|----------------|---------------------|
> | **MoFO**              | **47.7**   |       | 44.3                       |       | 65.7     | 42.7           | 24.6                |
> | Vanilla-SoftMask      | 46.4       |       | 43.9                       |       | 65.6         | 42.9       | 23.2                |
> | CoFiTune w/o SoftMask | 37.7       |       | **44.4**                   |       | 65.4         | 42.1           | 25.8            |
> | CoFiTune w/ SoftMask  | 34.4       |       | 44.0                       |       | 65.0         | 41.5           | 25.6                |
>
> From the results, we can see that MoFO achieves higher scores on the fine-tuning tasks while effectively reducing knowledge forgetting, demonstrating its superiority over these methods.
>
> **References:**
>
> [1] Lin, Yong, et al. "Mitigating the alignment tax of rlhf." Proceedings of the 2024 Conference on Empirical Methods in Natural Language Processing. 2024.
>
> [2] Zhang, Hengyuan, et al. "Balancing speciality and versatility: a coarse to fine framework for supervised fine-tuning large language model." arXiv preprint arXiv:2404.10306 (2024).
>
> [3] Ke, Zixuan, et al. "Continual pre-training of language models." arXiv preprint arXiv:2302.03241 (2023).

---

> ### Author Response · Authors · 2024-11-25
> **Response (Part III)**
>
> ### **Response to Question 2**
>
> **(B). Combining Representitive Orthogonal Methods**
>
> **Clarification**: In the original manuscript, we study the combination of MoFO with two orthogonal methods GEM and Replay. We did NOT study the combination of MoFO and HFT, as they are both optimization-based methods and hence not orthogonal (see Section 2 in the revised manuscript for detialed categorization of different methods). In the revised manuscript, we also add the combination study of MoFO and EWC, a representative regularization-based method.
>
> **Justification of Representativeness**
>
> - GEM [4] is a well-established method that combines elements of both replay-based and optimization-based techniques [5][6]. It has been widely recognized in the continual learning community, as evidenced by its citation count of nearly 3000.
> - Replay is a widely adopted replay-based strategy [7] for mitigating forgetting, often employed as a baseline in recent studies [8][9].
> - EWC [10] is one of the pioneering regularization-based methods, detailed in Part \(C) of this response below. It is a highly important method in the continual learning field, with over 8,000 citations.
>
> We believe that these well-known methods would be representative enough to suggest the effectiveness of combining MoFO with orthoginal strategies.
>
> **References:**
>
> [4] Lopez-Paz, David, and Marc'Aurelio Ranzato. "Gradient episodic memory for continual learning." Advances in neural information processing systems 30 (2017).
>
> [5] Wang, Liyuan, et al. "A comprehensive survey of continual learning: theory, method and application." IEEE Transactions on Pattern Analysis and Machine Intelligence (2024).
>
> [6] De Lange, Matthias, et al. "A continual learning survey: Defying forgetting in classification tasks." IEEE transactions on pattern analysis and machine intelligence 44.7 (2021): 3366-3385.
>
> [7] Rolnick, David, et al. "Experience replay for continual learning." Advances in neural information processing systems 32 (2019).
>
> [8] Wang, Xiao, et al. "Orthogonal subspace learning for language model continual learning." arXiv preprint arXiv:2310.14152 (2023).
>
> [9] Wang, Xiao, et al. "TRACE: A Comprehensive Benchmark for Continual Learning in Large Language Models." arXiv preprint arXiv:2310.06762 (2023).
>
> [10] Kirkpatrick, James, et al. "Overcoming catastrophic forgetting in neural networks." Proceedings of the national academy of sciences 114.13 (2017): 3521-3526.

---

### Official Review · Reviewer_nhv1 · 2024-11-05

**Soundness:** 3
**Presentation:** 3
**Contribution:** 3
**Rating:** 6
**Confidence:** 3

**Summary:**

The paper presents a novel fine-tuning algorithm named Momentum-Filtered Optimizer (MoFO). The primary contribution of the paper is addressing the issue of catastrophic forgetting in LLMs during fine-tuning. MoFO selectively updates model parameters with the largest momentum magnitudes to maintain proximity to the pre-trained state, thereby mitigating knowledge forgetting. The authors claim that MoFO achieves similar performance to the default fine-tuning algorithm while effectively preserving pre-training knowledge, without requiring access to pre-training data. The paper is rigorous in its convergence analysis and experiments, demonstrating a good trade-off in maintaining general capabilities and downstream task performance.

**Strengths:**

1. This paper provides a comprehensive and reasonable proof of the convergence.
2. The effectiveness of the method is verified with LLMs of different sizes.
3. The convergence direction of MoFO and baselines is visualized using loss landscape, proving that MoFO converges to a closer point.

**Weaknesses:**

In Section 4, the loss landscapes for HFT and LoRA were not reported.

**Questions:**

1. The fraction of MoFO was varied in the experiments. Appendix E.1 only discusses its impact on the Llama-2-7B model. How does this hyperparameter affect models of different sizes?

2. MoFO updates only a small subset of parameters at each iteration, and the learning rate decreases over time. Will this lead to reduced fine-tuning efficiency?

---

> ### Author Response · Authors · 2024-11-25
>
> ### **Weakness 1**:
> *In Section 4, the loss landscapes for HFT and LoRA were not reported.*
>
> ### **Response to Weakness 1:**
> Thank you for pointing this out. We have updated Figure 5 in Section 5 in our revised manuscript to include the training paths of HFT and LoRA. The revised figure illustrates that both HFT and LoRA do not converge to minima as close to the pre-trained model as MoFO does, indicating that they may experience higher levels of forgetting compared to MoFO.
>
> We note that HFT optimizes in a manner similar to Random BCD.
>
> For LoRA, we make the following modelling in the landscape visualization example. The core principle of LoRA (Low-Rank Adaptation) involves approximating the original training space by a low-rank subspace. Since we consider a two-dimensional training space for visualizing the landscape, we set the rank of LoRA space to 1. Specifically, the parameters $\theta_1$ and $\theta_2$ exhibit a linear relationship. Given that the pretrained model is (0, 0), the parameters under LoRA are set to satisfy $\theta_2 = \beta \theta_1$. In Figure 5, we selected $\beta = 0.5$, and we see that LoRA converges to a closer local minimum than Default FT.
>
>
> ---
> ### **Question 1**
> *The fraction of MoFO was varied in the experiments. Appendix E.1 only discusses its impact on the Llama-2-7B model. How does this hyperparameter affect models of different sizes?*
>
> ### **Response to Question 1:**
>  Thanks for your suggestions, in our updated version, we fine-tune LLMs of various sizes (1B, 3B, 7B) with different fraction ratios ($\alpha$% = 5%, 10%, 20%, 40%, 80%). The results of this experiment are presented in Figure 6 and Figure 7 of Appendix E.1 in our revised manuscript, and we make the following key observations.
>
> **Key Observations:**
>
> a. **Good Fine-Tuning Performance at $\alpha$% $\approx$ 20%:** Across models of different sizes, setting the fraction $\alpha$% to approximately 20% allows MoFO to reach fine-tuning performance similar to the default FT (with up to 3% performance drop). Therefore, we recommend initiating with a 20\% update fraction when applying MoFO on new LLMs or fine-tuning tasks.
>
> b. **Enhanced Knowledge Retention in Larger Models:** Larger models tend to better retain the pre-training knowledge with MoFO. Specifically, for a given fraction ratio, MoFO provides more significant performance gains in mitigating forgetting as the model grows larger.
>
> ---
> ### **Question 2**
> *MoFO updates only a small subset of parameters at each iteration, and the learning rate decreases over time. Will this lead to reduced fine-tuning efficiency?*
>
>
> ### **Response to Question 2:**
>
> First, we would like to clarify that default FT (Adam) share **the same learning rate schedule**. Both methods use the default cosine learning rate schedule [1].
>
> Second, we find that MoFO does NOT lead to reduced fine-tuning efficiency. We compare the fine-tuning process and total training time between MoFO and default FT, as described below.
>
> (a). **Fine-tuning process:** We fine-tune the LLaMA2-7B model on the MetaMathQA dataset and evaluate the fine-tuning performance on the GSM8K benchmark during the training process. The results are presented in Figure 9 in Appendix E.4 of our revised manuscript. MoFO and default FT achieve comparable performance across the whole training process, demonstrating that **updating only a subset of parameters within each iteration in MoFO does not adversely affect the convergence speed or the final performance.**
>
> (b). **Total training time:** Although MoFO requires computing a mask to select parameters with the largest momentum magnitudes, the additional computational overhead is minor. Following the setting in Section 4.2 in the revised manuscript, we measured the training times for MoFO and default FT on different model sizes and datasets. The results are as follows:
>
> | Model                        | Default FT    | MoFO ($\alpha$% = 10%)             | Additional Training Time    |
> |------------------------------|-----------------------------|---------------------------|----------|
> | LLaMA3.2-1B                  | 49m22s       | 51m24s     | 4.1% |
> | LLaMA3.2-3B                  | 1h30m18s| 1h34m24s | 4.5% |
> | LLaMA3-8B on UltraFeedback dataset[2] | 5h2m0.69s | 5h17m34.01s | 5.0%  |
>
> As shown in the table, the additional training time incurred by MoFO is approximately 4–5%, which is relatively minor and manageable in practical applications.
>
> **Reference:**
>
> [1] Loshchilov, Ilya, and Frank Hutter. "Sgdr: Stochastic gradient descent with warm restarts." arXiv preprint arXiv:1608.03983 (2016).
>
> [2] Cui, Ganqu, et al. "ULTRAFEEDBACK: Boosting Language Models with Scaled AI Feedback." Forty-first International Conference on Machine Learning. 2024.

---

> ### Author Response · Authors · 2024-12-01
> **Seeking for your Valuable Feedback**
>
> Dear Reviewer,
>
> Thank you for the time and effort you have dedicated to reviewing our paper. We sincerely appreciate your insightful comments and suggestions. We have conducted additional experiments and analyses to address your concerns and questions in the responses above. Please feel free to review them at your convenience.
>
> Your comments have been highly valuable to us, and we have put much effort into running new experiments to address your concern. We would love to receive feedback from you. If your concern is addressed, we humbly invite the reviewer to consider increasing the score. Your support is deeply appreciated!

---

### Meta-Review · Area_Chair_jjzh · 2024-12-13

**Metareview:**

I have read all the materials of this paper including the manuscript, appendix, comments, and response. Based on collected information from all reviewers and my personal judgment, I can make the recommendation on this paper, reject. No objection from reviewers who participated in the internal discussion was raised against the reject recommendation.

**Research Question**

This paper considers the forgetting problem of LLM fine-tuning, i.e., the fine-tuned model performs worse on the original general tasks.

**Challenge Analysis**

The authors argue that two types of existing solutions need to access the original pre-training data or gradients, which are impractical in some scenarios.

**Philosophy**

The authors aim to solely employ the well-trained LLM model to tackle the above forgetting problem. To achieve this, the authors provide some empirical analysis, demonstrating the fine-tuned model performs well in the forgetting problem if the fine-tuned model is close to the original model. Thus, the authors aim to just update some partial parameters during the fine-tuning process. Here, the logic is not smooth. Moreover, the authors fail to verify their method can lead to a close-to-original model. (Major concern I)

**Technique**

The authors employ BCD and select the largest gradients to update. I am a little confused why updating largest gradients leads to a close model.  (Major concern II)

**Theoretical Analysis**

The authors demonstrate the convergence of the proposed new optimizer.

**Experiments**

1.	The experimental results seem promising.
2.	The Pareto analysis is a plus.
3.	However, the parameter $\alpha$ varies on different datasets. The authors need to provide how to set $\alpha$ in practice. (Major concern III; this is the key reason for my rejection recommendation.)
4. Again, the authors need to demonstrate the fine-tuned model by the proposed method is close to the original model. Appendix E2 is a supplementary evidence, but it is not enough, where all baseline methods should be included.


**Presentation**

1.	Table 3 can be improved in terms of visualization.
2.	Section 5 can be moved forward.

In general, this paper is very close to the bar. I would like to see an improved version in the future conference.

**Additional Comments On Reviewer Discussion:**

No objection from reviewers who participated in the internal discussion was raised against the reject recommendation.

---

### Decision · Program_Chairs · 2025-01-22

Reject